# On Fast Leverage Score Sampling and Optimal Learning

**Alessandro Rudi**[*]
INRIA – Sierra team,
ENS, Paris

**Daniele Calandriello**[*]
LCSL – IIT & MIT,
Genoa, Italy

**Luigi Carratino**
University of Genoa,
Genoa, Italy

**Lorenzo Rosasco**
University of Genoa,
LCSL – IIT & MIT

## Abstract

Leverage score sampling provides an appealing way to perform approximate computations for large matrices. Indeed, it allows to derive faithful approximations with a complexity adapted to the problem at hand. Yet, performing leverage scores sampling is a challenge in its own right requiring further approximations. In this paper, we study the problem of leverage score sampling for positive definite matrices defined by a kernel. Our contribution is twofold. First we provide a novel algorithm for leverage score sampling and second, we exploit the proposed method in statistical learning by deriving a novel solver for kernel ridge regression. Our main technical contribution is showing that the proposed algorithms are currently the most efficient and accurate for these problems.

## 1 Introduction

A variety of machine learning problems require manipulating and performing computations with large matrices that often do not fit memory. In practice, randomized techniques are often employed to reduce the computational burden. Examples include stochastic approximations [1], columns/rows subsampling and more general sketching techniques [2, 3]. One of the simplest approach is uniform column sampling [4, 5], that is replacing the original matrix with a subset of columns chosen uniformly at random. This approach is fast to compute, but the number of columns needed for a prescribed approximation accuracy does not take advantage of the possible low rank structure of the matrix at hand. As discussed in [6], leverage score sampling provides a way to tackle this shortcoming. Here columns are sampled proportionally to suitable weights, called leverage scores (LS) [7, 6]. With this sampling strategy, the number of columns needed for a prescribed accuracy is governed by the so called *effective dimension* which is a natural extension of the notion of rank. Despite these nice properties, performing leverage score sampling provides a challenge in its own right, since it has complexity in the same order of an eigendecomposition of the original matrix. Indeed, much effort has been recently devoted to derive fast and provably accurate algorithms for approximate leverage score sampling [2, 8, 6, 9, 10].

In this paper, we consider these questions in the case of positive semi-definite matrices, central for example in Gaussian processes [11] and kernel methods [12]. Sampling approaches in this context are related to the so called Nyström approximation [13] and Nyström centers selection problem [11], and are widely studied both in practice [4] and in theory [5]. Our contribution is twofold. First, we propose and study BLESS, a novel algorithm for approximate leverage scores sampling. The first solution to this problem is introduced in [6], but has poor approximation guarantees and high time complexity. Improved approximations are achieved by algorithms recently proposed in [8] and [9]. In particular, the approach in [8] can obtain good accuracy and very efficient computations but only as long as distributed resources are available. Our first technical contribution is showing that our algorithm can achieve state of the art accuracy and computational complexity without requiring

---

[*]Equal contribution. Respective emails: alessandro.rudi@inria.fr, daniele.calandriello@iit.it

distributed resources. The key idea is to follow a coarse to fine strategy, alternating uniform and leverage scores sampling on sets of increasing size.

Our second, contribution is considering leverage score sampling in statistical learning with least squares. We extend the approach in [14] for efficient kernel ridge regression based on combining fast optimization algorithms (preconditioned conjugate gradient) with uniform sampling. Results in [14] showed that optimal learning bounds can be achieved with a complexity which is $\widetilde{\mathcal{O}}(n\sqrt{n})$ in time and $\widetilde{\mathcal{O}}(n)$ space. In this paper, we study the impact of replacing uniform with leverage score sampling. In particular, we prove that the derived method still achieves optimal learning bounds but the time and memory is now $\widetilde{\mathcal{O}}(nd_{\mathrm{eff}})$, and $\widetilde{\mathcal{O}}(d_{\mathrm{eff}}{}^2)$ respectively, where $d_{\mathrm{eff}}$ is the effective dimension which and is never larger, and possibly much smaller, than $\sqrt{n}$. To the best of our knowledge this is the best currently known computational guarantees for a kernel ridge regression solver.

## 2 Leverage score sampling with BLESS

After introducing leverage score sampling and previous algorithms, we present our approach and first theoretical results.

### 2.1 Leverage score sampling

Suppose $\widehat{K} \in \mathbb{R}^{n \times n}$ is symmetric and positive semidefinite. A basic question is deriving memory efficient approximation of $\widehat{K}$ [4, 8] or related quantities, e.g. approximate projections on its range [9], or associated estimators, as in kernel ridge regression [15, 14]. The eigendecomposition of $\widehat{K}$ offers a natural, but computationally demanding solution. Subsampling columns (or rows) is an appealing alternative. A basic approach is uniform sampling, whereas a more refined approach is leverage scores sampling. This latter procedure corresponds to sampling columns with probabilities proportional to the leverage scores

$$\ell(i, \lambda) = \left( \widehat{K}(\widehat{K} + \lambda n I)^{-1} \right)_{ii}, \qquad i \in [n], \tag{1}$$

where $[n] = \{1, \ldots, n\}$. The advantage of leverage score sampling, is that potentially very few columns can suffice for the desired approximation. Indeed, letting

$$d_\infty(\lambda) = n \max_{i=1,\ldots,n} \ell(i, \lambda), \qquad d_{\mathrm{eff}}(\lambda) = \sum_{i=1}^{n} \ell(i, \lambda),$$

for $\lambda > 0$, it is easy to see that $d_{\mathrm{eff}}(\lambda) \leq d_\infty(\lambda) \leq 1/\lambda$ for all $\lambda$, and previous results show that the number of columns required for accurate approximation are $d_\infty$ for uniform sampling and $d_{\mathrm{eff}}$ for leverage score sampling [5, 6]. However, it is clear from definition (1) that an exact leverage scores computation would require the same order of computations as an eigendecomposition, hence approximations are needed. The accuracy of approximate leverage scores is typically measured by $t > 0$ in multiplicative bounds of the form

$$\frac{1}{1+t} \ell(i, \lambda) \leq \widetilde{\ell}(i, \lambda) \leq (1+t)\ell(i, \lambda), \quad \forall i \in [n]. \tag{2}$$

Before proposing a new improved solution, we briefly discuss relevant previous works. To provide a unified view, some preliminary discussion is useful.

### 2.2 Approximate leverage scores

First, we recall how a subset of columns can be used to compute approximate leverage scores. For $M \leq n$, let $J = \{j_i\}_{i=1}^{M}$ with $j_i \in [n]$, and $\widehat{K}_{J,J} \in \mathbb{R}^{M \times M}$ with entries $(K_{J,J})_{lm} = K_{j_l, j_m}$. For $i \in [n]$, let $\widehat{K}_{J,i} = (\widehat{K}_{j_1,i}, \ldots, \widehat{K}_{j_M,i})$ and consider for $\lambda > 1/n$,

$$\widetilde{\ell}_J(i, \lambda) = (\lambda n)^{-1}(\widehat{K}_{ii} - \widehat{K}_{J,i}^\top (\widehat{K}_{J,J} + \lambda n A)^{-1} \widehat{K}_{J,i}), \tag{3}$$

where $A \in \mathbb{R}^{M \times M}$ is a matrix to be specified [*] (see later for details). The above definition is motivated by the observation that if $J = [n]$, and $A = I$, then $\widetilde{\ell}_J(i, \lambda) = \ell(i, \lambda)$, by the following

identity

$$\widehat{K}(\widehat{K} + \lambda nI)^{-1} = (\lambda n)^{-1}(\widehat{K} - \widehat{K}(\widehat{K} + \lambda nI)^{-1}\widehat{K}).$$

In the following, it is also useful to consider a subset of leverage scores computed as in (3). For $M \le R \le n$, let $U = \{u_i\}_{i=1}^R$ with $u_i \in [n]$, and

$$L_J(U, \lambda) = \{\widetilde{\ell}_J(u_1, \lambda), \ldots, \widetilde{\ell}_J(u_R, \lambda)\}. \tag{4}$$

Also in the following we will use the notation

$$L_J(U, \lambda) \mapsto J' \tag{5}$$

to indicate the leverage score sampling of $J' \subset U$ columns based on the leverage scores $L_J(U, \lambda)$, that is the procedure of sampling columns from $U$ according to their leverage scores 1, computed using $J$, to obtain a new subset of columns $J'$.

We end noting that leverage score sampling (5) requires $\mathcal{O}(M^2)$ memory to store $K_J$, and $\mathcal{O}(M^3 + RM^2)$ time to invert $K_J$, and compute $R$ leverage scores via (3).

## 2.3 Previous algorithms for leverage scores computations

We discuss relevant previous approaches using the above quantities.

TWO-PASS sampling [6]. This is the first approximate leverage score sampling proposed, and is based on using directly (5) as $L_{J_1}(U_2, \lambda) \mapsto J_2$, with $U_2 = [n]$ and $J_1$ a subset taken uniformly at random. Here we call this method TWO-PASS sampling since it requires two rounds of sampling on the whole set $[n]$, one uniform to select $J_1$ and one using leverage scores to select $J_2$.

RECURSIVE-RLS [9]. This is a development of TWO-PASS sampling based on the idea of recursing the above construction. In our notation, let $U_1 \subset U_2 \subset U_3 = [n]$, where $U_1, U_2$ are uniformly sampled and have cardinalities $n/4$ and $n/2$, respectively. The idea is to start from $J_1 = U_1$, and consider first

$$L_{J_1}(U_2, \lambda) \mapsto J_2,$$

but then continue with

$$L_{J_2}(U_3, \lambda) \mapsto J_3.$$

Indeed, the above construction can be made recursive for a family of nested subsets $(U_h)_H$ of cardinalities $n/2^h$, considering $J_1 = U_1$ and

$$L_{J_h}(U_{h+1}, \lambda) \mapsto J_{h+1}. \tag{6}$$

SQUEAK[8]. This approach follows a different iterative strategy. Consider a partition $U_1, U_2, U_3$ of $[n]$, so that $U_j = n/3$, for $j = 1, \ldots 3$. Then, consider $J_1 = U_1$, and

$$L_{J_1 \cup U_2}(J_1 \cup U_2, \lambda) \mapsto J_2,$$

and then continue with

$$L_{J_2 \cup U_3}(J_2 \cup U_3, \lambda) \mapsto J_3.$$

Similarly to the other cases, the procedure is iterated considering $H$ subsets $(U_h)_{h=1}^H$ each with cardinality $n/H$. Starting from $J_1 = U_1$ the iterations is

$$L_{J_h \cup U_{h+1}}(J_h \cup U_{h+1}, \lambda). \tag{7}$$

We note that all the above procedures require specifying the number of iteration to be performed, the weights matrix to compute the leverage scores at each iteration, and a strategy to select the subsets $(U_h)_h$. In all the above cases the selection of $U_h$ is based on uniform sampling, while the number of iterations and weight choices arise from theoretical considerations (see [6, 8, 9] for details).

Note that TWO-PASS SAMPLING uses a set $J_1$ of cardinality roughly $1/\lambda$ (an upper bound on $d_\infty(\lambda)$) and incurs in a computational cost of $RM^2 = n/\lambda^2$. In comparison, RECURSIVE-RLS [9] leads to essentially the same accuracy while improving computations. In particular, the sets $J_h$ are never larger than $d_{\text{eff}}(\lambda)$. Taking into account that at the last iteration performs leverage score sampling on $U_h = [n]$, the total computational complexity is $nd_{\text{eff}}(\lambda)^2$. SQUEAK [8] recovers the same accuracy, size of $J_h$, and $nd_{\text{eff}}(\lambda)^2$ time complexity when $|U_h| \simeq d_{\text{eff}}(\lambda)$, but only requires a single pass over the data. We also note that a distributed version of SQUEAK is discussed in [8], which allows to reduce the computational cost to $nd_{\text{eff}}(\lambda)^2/p$, provided $p$ machines are available.

**Algorithm 1** Bottom-up Leverage Scores Sampling (BLESS)

---

**Input:** dataset $\{x_i\}_{i=1}^n$, regularization $\lambda$, step $q$, starting reg. $\lambda_0$, constants $q_1, q_2$ controlling the approximation level.

**Output:** $M_h \in [n]$ number of selected points, $J_h$ set of indexes, $A_h$ weights.

1:   $J_0 = \emptyset$, $A_0 = []$, $H = \frac{\log(\lambda_0/\lambda)}{\log q}$
2: **for** $h = 1 \ldots H$ **do**
3:     $\lambda_h = \lambda_{h-1}/q$
4:     set constant $R_h = q_1 \min\{\kappa^2/\lambda_h, \ n\}$
5:     sample $U_h = \{u_1, \ldots, u_{R_h}\}$ i.i.d. $u_i \sim Uniform([n])$
6:     compute $\widetilde{\ell}_{J_{h-1}}(x_{u_k}, \lambda_h)$ for all $u_k \in U_h$ using Eq. 3
7:     set $P_h = (p_{h,k})_{k=1}^{R_h}$ with $p_{h,k} = \widetilde{\ell}_{J_{h-1}}(x_{u_k}, \lambda_h) / (\sum_{u \in U_h} \widetilde{\ell}_{J_{h-1}}(x_u, \lambda_h))$
8:     set constant $M_h = q_2 d_h$ with $d_h = \frac{n}{R_h} \sum_{u \in U_h} \widetilde{\ell}_{J_{h-1}}(x_u, \lambda_h)$, and
9:     sample $J_h = \{j_1, \ldots, j_{M_h}\}$ i.i.d. $j_i \sim Multinomial(P_h, U_h)$
10:     $A_h = \frac{R_h M_h}{n} \text{diag}\left(p_{h,j_1}, \ldots, p_{h,j_{M_h}}\right)$
11: **end for**

---

## 2.4 Leverage score sampling with BLESS

The procedure we propose, dubbed BLESS, has similarities to the one proposed in [9] (see (6)), but also some important differences. The main difference is that, rather than a fixed $\lambda$, we consider a decreasing sequence of parameters $\lambda_0 > \lambda_1 > \cdots > \lambda_H = \lambda$ resulting in different algorithmic choices. For the construction of the subsets $U_h$ we do not use nested subsets, but rather each $(U_h)_{h=1}^H$ is sampled uniformly and independently, with a size smoothly increasing as $1/\lambda_h$. Similarly, as in [9] we proceed iteratively, but at each iteration a different decreasing parameter $\lambda_h$ is used to compute the leverage scores. Using the notation introduced above, the iteration of BLESS is given by

$$L_{J_h}(U_{h+1}, \lambda_{h+1}) \mapsto J_{h+1}, \tag{8}$$

where the initial set $J_1 = U_1$ is sampled uniformly with size roughly $1/\lambda_0$.

BLESS has two main advantages. The first is computational: each of the sets $U_h$, including the final $U_H$, has cardinality smaller than $1/\lambda$. Therefore the overall runtime has a cost of only $RM^2 \leq M^2/\lambda$, which can be dramatically smaller than the $nM^2$ cost achieved by the methods in [9], [8] and is comparable to the distributed version of SQUEAK using $p = \lambda/n$ machines. The second advantage is that a whole *path* of leverage scores $\{\ell(i, \lambda_h)\}_{h=1}^H$ is computed at once, in the sense that at each iteration accurate approximate leverage scores at scale $\lambda_h$ are computed. This is extremely useful in practice, as it can be used when cross-validating $\lambda_h$. As a comparison, for all previous method a full run of the algorithm is needed for each value of $\lambda_h$.

In the paper we consider two variations of the above general idea leading to Algorithm 1 and Algorithm 2. The main difference in the two algorithms lies in the way in which sampling is performed: with and without replacement, respectively. In particular, considering sampling without replacement (see 2) it is possible to take the set $(U_h)_{h=1}^H$ to be nested and also to obtain slightly improved results, as shown in the next section.

The derivation of BLESS rests on some basic ideas. First, note that, since sampling uniformly a set $U_\lambda$ of size $d_\infty(\lambda) \leq 1/\lambda$ allows a good approximation, then we can replace $L_{[n]}([n], \lambda) \mapsto J$ by

$$L_{U_\lambda}(U_\lambda, \lambda) \mapsto J, \tag{9}$$

where $J$ can be taken to have cardinality $d_{\text{eff}}(\lambda)$. However, this is still costly, and the idea is to repeat and couple approximations at multiple scales. Consider $\lambda' > \lambda$, a set $U_{\lambda'}$ of size $d_\infty(\lambda') \leq 1/\lambda'$ sampled uniformly, and $L_{U_{\lambda'}}(U_{\lambda'}, \lambda') \mapsto J'$. The basic idea behind BLESS is to replace (9) by

$$L_{J'}(U_\lambda, \lambda) \mapsto \tilde{J}.$$

The key result, see , is that taking $\tilde{J}$ of cardinality

$$(\lambda'/\lambda) d_{\text{eff}}(\lambda) \tag{10}$$

suffice to achieve the same accuracy as $J$. Now, if we take $\lambda'$ sufficiently large, it is easy to see that $d_{\text{eff}}(\lambda') \sim d_\infty(\lambda') \sim 1/\lambda'$, so that we can take $J'$ uniformly at random. However, the factor $(\lambda'/\lambda)$ in (10) becomes too big. Taking multiple scales fix this problem and leads to the iteration in (8).

**Algorithm 2** Bottom-up Leverage Scores Sampling without Replacement (BLESS-R)

**Input:** dataset $\{x_i\}_{i=1}^n$, regularization $\lambda$, step $q$, starting reg. $\lambda_0$, constant $q_2$ controlling the approximation level.

**Output:** $M_h \in [n]$ number of selected points, $J_h$ set of indexes, $A_h$ weights.

1: $J_0 = \emptyset$, $A_0 = []$, $H = \frac{\log(\lambda_0/\lambda)}{\log q}$,
2: **for** $h = 1 \ldots H$ **do**
3:     $\lambda_h = \lambda_{h-1}/q$
4:     set constant $\beta_h = \min\{q_2\kappa^2/(\lambda_h n), 1\}$
5:     initialize $U_h = \emptyset$
6:     **for** $i \in [n]$ **do**
7:         add $i$ to $U_h$ with probability $\beta_h$
8:     **end for**
9:     **for** $j \in U_h$ **do**
10:        compute $p_{h,j} = \min\{q_2\widetilde{\ell}_{J_{h-1}}(x_j, \lambda_{h-1}), 1\}$
11:        add $j$ to $J_h$ with probability $p_{h,j}/\beta_h$
12:     **end for**
13:     $J_h = \{j_1, \ldots, j_{M_h}\}$, and $A_h = \mathrm{diag}\left(p_{h,j_1}, \ldots, p_{h,j_{M_h}}\right)$.
14: **end for**

## 2.5 Theoretical guarantees

Our first main result establishes in a precise and quantitative way the advantages of BLESS.

**Theorem 1.** *Let $n \in \mathbb{N}$, $\lambda > 0$ and $\delta \in (0, 1]$. Given $t > 0, q > 1$ and $H \in \mathbb{N}$, $(\lambda_h)_{h=1}^H$ defined as in Algorithms 1 and 2, when $(J_h, a_h)_{h=1}^H$ are computed*

    *1. by Alg. 1 with parameters $\lambda_0 = \frac{\kappa^2}{\min(t,1)}$, $q_1 \geq \frac{5\kappa^2 q_2}{q(1+t)}$, $q_2 \geq 12q\frac{(2t+1)^2}{t^2}(1+t)\log\frac{12Hn}{\delta}$,*

    *2. by Alg. 2 with parameters $\lambda_0 = \frac{\kappa^2}{\min(t,1)}$, $q_1 \geq 54\kappa^2\frac{(2t+1)^2}{t^2}\log\frac{12Hn}{\delta}$,*

*let $\widetilde{\ell}_{J_h}(i, \lambda_h)$ as in Eq. (3) depending on $J_h, A_h$, then with probability at least $1 - \delta$:*

    *(a)*      $\frac{1}{1+t}\ell(i, \lambda_h) \leq \widetilde{\ell}_{J_h}(i, \lambda_h) \leq (1 + \min(t,1))\ell(i, \lambda_h), \quad \forall i \in [n], h \in [H],$

    *(b)*      $|J_h| \leq q_2 d_{\text{eff}}(\lambda_h), \quad \forall h \in [H].$

The above result confirms that the subsets $J_h$ computed by BLESS are accurate in the desired sense, see (2), and the size of all $J_h$ is small and proportional to $d_{\text{eff}}(\lambda_h)$, leading to a computational cost of only $\mathcal{O}\left(\min\left(\frac{1}{\lambda}, n\right) d_{\text{eff}}(\lambda)^2 \log^2 \frac{1}{\lambda}\right)$ in time and $O\left(d_{\text{eff}}(\lambda)^2 \log^2 \frac{1}{\lambda}\right)$ in space (for additional properties of $J_h$ see Thm. 4 in appendixes). Table 1 compares the complexity and number of columns sampled by BLESS with other methods. The crucial point is that in most applications, the parameter $\lambda$ is chosen as a decreasing function of $n$, e.g. $\lambda = 1/\sqrt{n}$, resulting in potentially massive computational gains. Indeed, since BLESS computes leverage scores for sets of size at most $1/\lambda$, this allows to perform leverage scores sampling on matrices with millions of rows/columns, as shown in the experiments. In the next section, we illustrate the impact of BLESS in the context of supervised statistical learning.

## 3 Efficient supervised learning with leverage scores

In this section, we discuss the impact of BLESS in a supervised learning. Unlike most previous results on leverage scores sampling in this context [6, 8, 9], we consider the setting of statistical learning, where the challenge is that inputs, as well as the outputs, are random. More precisely, given a probability space $(X \times Y, \rho)$, where $Y \subset \mathbb{R}$, and considering least squares, the problem is to solve

$$\min_{f \in \mathcal{H}} \mathcal{E}(f), \quad \mathcal{E}(f) = \int_{X \times Y} (f(x) - y)^2 d\rho(x, y), \tag{11}$$

| Algorithm | Runtime | $|J|$ |
|---|---|---|
| Uniform Sampling [5] | $-$ | $1/\lambda$ |
| Exact RLS Sampl. | $n^3$ | $d_{\text{eff}}(\lambda)$ |
| Two-Pass Sampling [6] | $n/\lambda^2$ | $d_{\text{eff}}(\lambda)$ |
| Recursive RLS [9] | $nd_{\text{eff}}(\lambda)^2$ | $d_{\text{eff}}(\lambda)$ |
| SQUEAK [8] | $nd_{\text{eff}}(\lambda)^2$ | $d_{\text{eff}}(\lambda)$ |
| This work, Alg. 1 and 2 | $\mathbf{1/\lambda\, d_{\text{eff}}(\lambda)^2}$ | $d_{\text{eff}}(\lambda)$ |

Table 1: The proposed algorithms are compared with the state of the art (in $\widetilde{\mathcal{O}}$ notation), in terms of time complexity and cardinality of the set $J$ required to satisfy the approximation condition in Eq. 2.

when $\rho$ is known only through $(x_i, y_i)_{i=1}^n \sim \rho^n$. In the above minimization problem, $\mathcal{H}$ is a reproducing kernel Hilbert space defined by a positive definite kernel $K : X \times X \to \mathbb{R}$ [12]. Recall that the latter is defined as the completion of $\text{span}\{K(x, \cdot) \mid x \in X\}$ with the inner product $\langle K(x, \cdot), K(x', \cdot)\rangle_{\mathcal{H}} = K(x, x')$. The quality of an empirical approximate solution $\widehat{f}$ is measured via probabilistic bounds on the excess risk $\mathcal{R}(\widehat{f}) = \mathcal{E}(\widehat{f}) - \min_{f \in \mathcal{H}} \mathcal{E}(f)$.

## 3.1 Learning with FALKON-BLESS

The algorithm we propose, called FALKON-BLESS, combines BLESS with FALKON [14] a state of the art algorithm to solve the least squares problem presented above. The appeal of FALKON is that it is currently the most efficient solution to achieve optimal excess risk bounds. As we discuss in the following, the combination with BLESS leads to further improvements.

We describe the derivation of the considered algorithm starting from kernel ridge regression (KRR)

$$\widehat{f}_\lambda(x) = \sum_{i=1}^n K(x, x_i)c_i, \qquad c = (\widehat{K} + \lambda n I)^{-1}\widehat{Y} \tag{12}$$

where $c = (c_1, \ldots, c_n)$, $\widehat{Y} = (y_1, \ldots, y_n)$ and $\widehat{K} \in \mathbb{R}^{n \times n}$ is the empirical kernel matrix with entries $(\widehat{K})_{ij} = K(x_i, x_j)$. KRR has optimal statistical properties [16], but large $\mathcal{O}(n^3)$ time and $\mathcal{O}(n^2)$ space requirements. FALKON can be seen as an approximate ridge regression solver combining a number of algorithmic ideas. First, sampling is used to select a subset $\{\widetilde{x}_1, \ldots, \widetilde{x}_M\}$ of the input data uniformly at random, and to define an approximate solution

$$\widehat{f}_{\lambda, M}(x) = \sum_{j=1}^M K(\widetilde{x}_j, x)\alpha_j, \qquad \alpha = (K_{nM}^\top K_{nM} + \lambda K_{MM})^{-1} K_{nM}^\top y, \tag{13}$$

where $\alpha = (\alpha_1, \ldots, \alpha_M)$, $K_{nM} \in \mathbb{R}^{n \times M}$, has entries $(K_{nM})_{ij} = K(x_i, \widetilde{x}_j)$ and $K_{MM} \in \mathbb{R}^{M \times M}$ has entries $(K_{MM})_{jj'} = K(\widetilde{x}_j, \widetilde{x}_{j'})$, with $i \in [n], j, j' \in [M]$. We note, that the linear system in (13) can be seen to obtained from the one in (12) by uniform column subsampling of the empirical kernel matrix. The columns selected corresponds to the inputs $\{\widetilde{x}_1, \ldots, \widetilde{x}_M\}$. FALKON proposes to compute a solution of the linear system 13 via a preconditioned iterative solver. The preconditioner is the core of the algorithm and is defined by a matrix $B$ such that

$$BB^\top = \left(\frac{n}{M} K_{MM}^2 + \lambda K_{MM}\right)^{-1}. \tag{14}$$

The above choice provides a computationally efficient approximation to the exact preconditioner of the linear system in (13) corresponding to $B$ such that $BB^\top = (K_{nM}^\top K_{nM} + \lambda K_{MM})^{-1}$. The preconditioner in (14) can then be combined with conjugate gradient to solve the linear system in (13). The overall algorithm has complexity $\mathcal{O}(nMt)$ in time and $\mathcal{O}(M^2)$ in space, where $t$ is the number of conjugate gradient iterations performed.

In this paper, we analyze a variation of FALKON where the points $\{\widetilde{x}_1, \ldots, \widetilde{x}_M\}$ are selected via leverage score sampling using BLESS, see Algorithm 1 or Algorithm 2, so that $M = M_h$ and $\widetilde{x}_k = x_{j_k}$, for $J_h = \{j_1, \ldots, j_{M_h}\}$ and $k \in [M_h]$. Further, the preconditioner in (14) is replaced by

$$B_h B_h^\top = \left(\frac{n}{M} K_{J_h, J_h} A_h^{-1} K_{J_h, J_h} + \lambda_h K_{J_h, J_h}\right)^{-1}. \tag{15}$$

|         | Time | R-ACC | $5^{th}/95^{th}$ quant |
|---------|------|-------|-----------------------|
| BLESS   | **17**  | 1.06  | 0.57 / 2.03 |
| BLESS-R | **17**  | 1.06  | 0.73 / 1.50 |
| SQUEAK  | 52   | 1.06  | 0.70 / 1.48 |
| Uniform | -    | 1.09  | 0.22 / 3.75 |
| RRLS    | 235  | 1.59  | 1.00 / 2.70 |

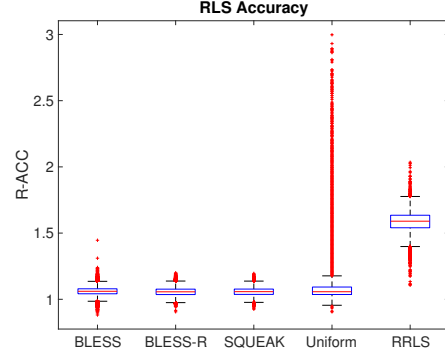

Figure 1: Leverage scores relative accuracy for $\lambda = 10^{-5}$, $n = 70\,000$, $M = 10\,000$, 10 repetitions.

This solution can lead to huge computational improvements. Indeed, the total cost of FALKON-BLESS is the sum of computing BLESS and FALKON, corresponding to

$$O\left(nMt + (1/\lambda)M^2\log n + M^3\right) \qquad \mathcal{O}(M^2), \tag{16}$$

in time and space respectively, where $M$ is the size of the set $J_H$ returned by BLESS.

### 3.2 Statistical properties of FALKON-BLESS

In this section, we state and discuss our second main result, providing an excess risk bound for FALKON-BLESS. Here a population version of the effective dimension plays a key role. Let $\rho_X$ be the marginal measure of $\rho$ on $X$, let $C : \mathcal{H} \to \mathcal{H}$ be the linear operator defined as follows and $d_{\text{eff}}{}^*(\lambda)$ be the population version of $d_{\text{eff}}(\lambda)$,

$$d_{\text{eff}}{}^*(\lambda) = \text{Tr}(C(C + \lambda I)^{-1}), \quad \text{with} \quad (Cf)(x') = \int_X K(x', x)f(x)d\rho_X(x),$$

for any $f \in \mathcal{H}, x \in X$. It is possible to show that $d_{\text{eff}}{}^*(\lambda)$ is the limit of $d_{\text{eff}}(\lambda)$ as $n$ goes to infinity, see Lemma 1 below taken from [15]. If we assume throughout that,

$$K(x, x') \leq \kappa^2, \quad \forall x, x' \in X, \tag{17}$$

then the operator $C$ is symmetric, positive definite and trace class, and the behavior of $d_{\text{eff}}{}^*(\lambda)$ can be characterized in terms of the properties of the eigenvalues $(\sigma_j)_{j\in\mathbb{N}}$ of $C$. Indeed as for $d_{\text{eff}}(\lambda)$, we have that $d_{\text{eff}}{}^*(\lambda) \leq \kappa^2/\lambda$, moreover if $\sigma_j = \mathcal{O}(j^{-\alpha})$, for $\alpha \geq 1$, we have $d_{\text{eff}}{}^*(\lambda) = \mathcal{O}(\lambda^{-1/\alpha})$. Then for larger $\alpha$, $d_{\text{eff}}{}^*$ is smaller than $1/\lambda$ and faster learning rates are possible, as shown below. We next discuss the properties of the FALKON-BLESS solution denoted by $\widehat{f}_{\lambda,n,t}$.

**Theorem 2.** *Let $n \in \mathbb{N}$, $\lambda > 0$ and $\delta \in (0, 1]$. Assume that $y \in [-\frac{a}{2}, \frac{a}{2}]$, almost surely, $a > 0$, and denote by $f_\mathcal{H}$ a minimizer of (11). There exists $n_0 \in \mathbb{N}$, such that for any $n \geq n_0$, if $t \geq \log n$, $\lambda \geq \frac{9\kappa^2}{n}\log\frac{n}{\delta}$, then the following holds with probability at least $1 - \delta$:*

$$\mathcal{R}(\widehat{f}_{\lambda,n,t}) \leq \frac{4a}{n} + 32\|f_\mathcal{H}\|_\mathcal{H}^2 \left(\frac{a^2\log^2\frac{2}{\delta}}{n^2\lambda} + \frac{a\,d_{\text{eff}}(\lambda)\,\log\frac{2}{\delta}}{n} + \lambda\right).$$

*In particular, when $d_{\text{eff}}{}^*(\lambda) = \mathcal{O}(\lambda^{-1/\alpha})$, for $\alpha \geq 1$, by selecting $\lambda_* = n^{-\alpha/(\alpha+1)}$, we have*

$$\mathcal{R}(\widehat{f}_{\lambda_*,n,t}) \leq cn^{-\frac{\alpha}{\alpha+1}},$$

*where $c$ is given explicitly in the proof.*

We comment on the above result discussing the statistical and computational implications.

**Statistics.** The above theorem provides statistical guarantees in terms of finite sample bounds on the excess risk of FALKON-BLESS, A first bound depends of the number of examples $n$, the regularization parameter $\lambda$ and the population effective dimension $d_{\text{eff}}{}^*(\lambda)$. The second bound is derived optimizing $\lambda$, and is the same as the one achieved by exact kernel ridge regression which is

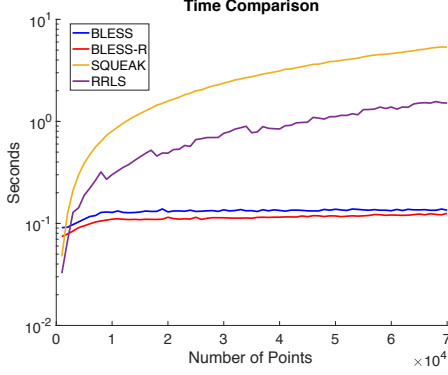
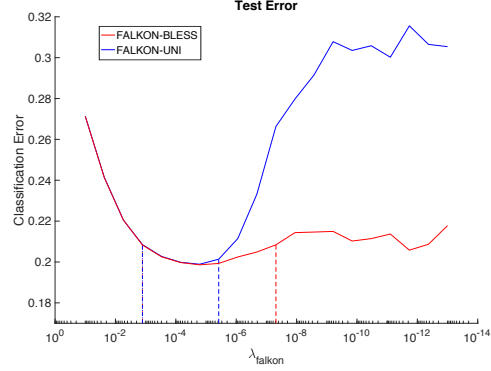

Figure 2: Runtimes with $\lambda = 10^{-3}$ and $n$ increasing    Figure 3: C-err at 5 iterations for varying $\lambda_{falkon}$

known to be optimal [16, 17, 18]. Note that improvements under further assumptions are possible and are derived in the supplementary materials, see Thm. 8. Here, we comment on the computational properties of FALKON-BLESS and compare it to previous solutions.

**Computations.** To discuss computational implications, we recall a result from [15] showing that the population version of the effective dimension $d_{\text{eff}}^*(\lambda)$ and the effective dimension $d_{\text{eff}}(\lambda)$ associated to the empirical kernel matrix converge up to constants.

**Lemma 1.** *Let $\lambda > 0$ and $\delta \in (0, 1]$. When $\lambda \geq \frac{9\kappa^2}{n} \log \frac{n}{\delta}$, then with probability at least $1 - \delta$,*

$$(1/3)d_{\text{eff}}^*(\lambda) \leq d_{\text{eff}}(\lambda) \leq 3d_{\text{eff}}^*(\lambda).$$

Recalling the complexity of FALKON-BLESS (16), using Thm 2 and Lemma 1, we derive a cost

$$\mathcal{O}\left( nd_{\text{eff}}^*(\lambda) \log n + \frac{1}{\lambda} d_{\text{eff}}^*(\lambda)^2 \log n + d_{\text{eff}}^*(\lambda)^3 \right)$$

in time and $\mathcal{O}(d_{\text{eff}}^*(\lambda)^2)$ in space, for all $n, \lambda$ satisfying the assumptions in Theorem 2. These expressions can be further simplified. Indeed, it is easy to see that for all $\lambda > 0$,

$$d_{\text{eff}}^*(\lambda) \leq \kappa^2/\lambda, \tag{18}$$

so that $d_{\text{eff}}^*(\lambda)^3 \leq \frac{\kappa^2}{\lambda} d_{\text{eff}}^*(\lambda)^2$. Moreover, if we consider the optimal choice $\lambda_* = \mathcal{O}(n^{-\frac{\alpha}{\alpha+1}})$ given in Theorem 2, and take $d_{\text{eff}}^*(\lambda) = \mathcal{O}(\lambda^{-1/\alpha})$, we have $\frac{1}{\lambda_*} d_{\text{eff}}^*(\lambda_*) \leq \mathcal{O}(n)$, and therefore $\frac{1}{\lambda} d_{\text{eff}}^*(\lambda)^2 \leq \mathcal{O}(nd_{\text{eff}}^*(\lambda))$. In summary, for the parameter choices leading to optimal learning rates, FALKON-BLESS has complexity $\widetilde{\mathcal{O}}(nd_{\text{eff}}^*(\lambda_*))$, in time and $\widetilde{\mathcal{O}}(d_{\text{eff}}^*(\lambda_*)^2)$ in space, ignoring log terms. We can compare this to previous results. In [14] uniform sampling is considered leading to $M \leq \mathcal{O}(1/\lambda)$ and achieving a complexity of $\widetilde{\mathcal{O}}(n/\lambda)$ which is always larger than the one achieved by FALKON in view of (18). Approximate leverage scores sampling is also considered in [14] requiring $\widetilde{\mathcal{O}}(nd_{\text{eff}}(\lambda)^2)$ time and reducing the time complexity of FALKON to $\widetilde{\mathcal{O}}(nd_{\text{eff}}(\lambda_*))$. Clearly in this case the complexity of leverage scores sampling dominates, and our results provide BLESS as a fix.

## 4 Experiments

**Leverage scores accuracy.** We first study the accuracy of the leverage scores generated by BLESS and BLESS-R, comparing SQUEAK [8] and Recursive-RLS (RRLS) [9]. We begin by uniformly sampling a subsets of $n = 7 \times 10^4$ points from the SUSY dataset [19], and computing the exact leverage scores $\ell(i, \lambda)$ using a Gaussian Kernel with $\sigma = 4$ and $\lambda = 10^{-5}$, which is at the limit of our computational feasibility. We then run each algorithm to compute the approximate leverage scores $\widetilde{\ell}_{J_H}(i, \lambda)$, and we measure the accuracy of each method using the ratio $\widetilde{\ell}_{J_H}(i, \lambda)/\ell(i, \lambda)$ (R-ACC). The final results are presented in Figure 1. On the left side for each algorithm we report runtime, mean R-ACC, and the $5^{th}$ and $95^{th}$ quantile, each averaged over the 10 repetitions. On the right side a box-plot of the R-ACC. As shown in Figure 1 BLESS and BLESS-R achieve the same optimal accuracy

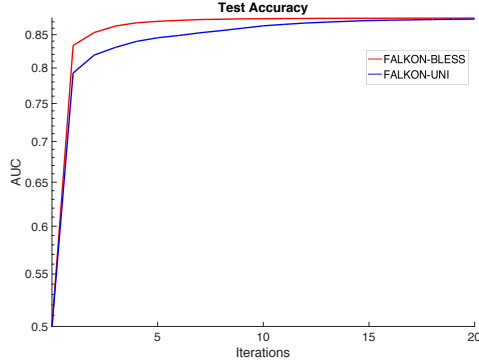
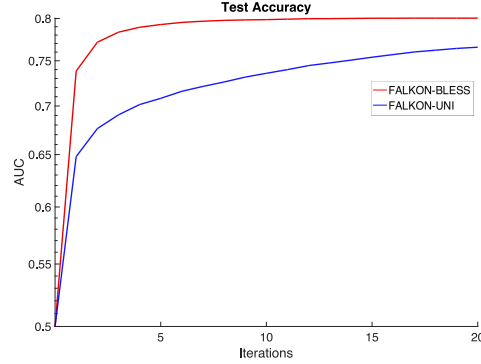

Figure 4: AUC per iteration of the SUSY dataset    Figure 5: AUC per iteration of the HIGGS dataset

of SQUEAK with just a fraction of time. Note that despite our best efforts, we could not obtain high-accuracy results for RRLS (maybe a wrong constant in the original implementation). However note that RRLS is computationally demanding compared to BLESS, being orders of magnitude slower, as expected from the theory. Finally, although uniform sampling is the fastest approach, it suffers from much larger variance and can over or under-estimate leverage scores by an order of magnitude more than the other methods, making it more fragile for downstream applications.

In Fig. 2 we plot the runtime cost of the compared algorithms as the number of points grows from $n = 1000$ to $70000$, this time for $\lambda = 10^{-3}$. We see that while previous algorithms' runtime grows near-linearly with $n$, BLESS and BLESS-R run in a constant $1/\lambda$ runtime, as predicted by the theory.

**BLESS for supervised learning.** We study the performance of FALKON-BLESS and compare it with the original FALKON [14] where an equal number of Nyström centres are sampled uniformly at random (FALKON-UNI). We take from [14] the two biggest datasets and their best hyper-parameters for the FALKON algorithm.

We noticed that it is possible to achieve the same accuracy of FALKON-UNI, by using $\lambda_{bless}$ for BLESS and $\lambda_{falkon}$ for FALKON with $\lambda_{bless} \gg \lambda_{falkon}$, in order to lower the $d_{\text{eff}}$ and keep the number of Nyström centres low. For the SUSY dataset we use a Gaussian Kernel with $\sigma = 4, \lambda_{falkon} = 10^{-6}, \lambda_{bless} = 10^{-4}$ obtaining $M_H \simeq 10^4$ Nyström centres. For the HIGGS dataset we use a Gaussian Kernel with $\sigma = 22, \lambda_{falkon} = 10^{-8}, \lambda_{bless} = 10^{-6}$, obtaining $M_H \simeq 3 \times 10^4$ Nyström centres. We then sample a comparable number of centers uniformly for FALKON-UNI. Looking at the plot of their AUC at each iteration (Fig. 4,5) we observe that FALKON-BLESS converges much faster than FALKON-UNI. For the SUSY dataset (Figure 4) 5 iterations of FALKON-BLESS (160 seconds) achieve the same accuracy of 20 iterations of FALKON-UNI (610 seconds). Since running BLESS takes just 12 secs. this corresponds to a $\sim 4\times$ speedup. For the HIGGS dataset 10 iter. of FALKON-BLESS (with BLESS requiring $1.5$ minutes, for a total of $1.4$ hours) achieve better accuracy of 20 iter. of FALKON-UNI ($2.7$ hours). Additionally we observed that FALKON-BLESS is more stable than FALKON-UNI w.r.t. $\lambda_{falkon}, \sigma$. In Figure 3 the classification error after 5 iterations of FALKON-BLESS and FALKON-UNI over the SUSY dataset ($\lambda_{bless} = 10^{-4}$). We notice that FALKON-BLESS has a wider optimal region ($95\%$ of the best error) for the regulariazion parameter ($[1.3 \times 10^{-3}, 4.8 \times 10^{-8}]$) w.r.t. FALKON-UNI ($[1.3 \times 10^{-3}, 3.8 \times 10^{-6}]$).

# 5    Conclusions

In this paper we presented two algorithms BLESS and BLESS-R to efficiently compute a small set of columns from a large symmetric positive semidefinite matrix $K$, useful for approximating the matrix or to compute leverage scores with a given precision. Moreover we applied the proposed algorithms in the context of statistical learning with least squares, combining BLESS with FALKON [14]. We analyzed the computational and statistical properties of the resulting algorithm, showing that it achieves optimal statistical guarantees with a cost that is $O(nd_{\text{eff}}^*(\lambda))$ in time, being currently the fastest. We can extend the proposed work in several ways: (a) combine BLESS with fast stochastic [20] or online [21] gradient algorithms and other approximation schemes (i.e. random features [22, 23, 24]), to further reduce the computational complexity for optimal rates, (b) consider the impact of BLESS in the context of multi-tasking [25, 26] or structured prediction [27, 28].

**Acknowledgments.**
This material is based upon work supported by the Center for Brains, Minds and Machines (CBMM), funded by NSF STC award CCF-1231216, and the Italian Institute of Technology. We gratefully acknowledge the support of NVIDIA Corporation for the donation of the Titan Xp GPUs and the Tesla k40 GPU used for this research. L. R. acknowledges the support of the AFOSR projects FA9550-17-1-0390 and BAA-AFRL-AFOSR-2016-0007 (European Office of Aerospace Research and Development), and the EU H2020-MSCA-RISE project NoMADS - DLV-777826. A. R. acknowledges the support of the European Research Council (grant SEQUOIA 724063).

## Footnotes

*Clearly, $\widetilde{\ell}_J$ depends on the choice of the matrix $A$, but we omit this dependence to simplify the notation.

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
