[Supplementary Material]

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

# A  Theoretical Analysis for Algorithms 1 and 2

In this section, Thm. 4 and Thm. 5 provide guarantees for the two methods, from which Thm. 1 is derived.

In particular in Section A.4 some important properties about (out-of-sample-)leverage scores, that will be used in the proofs, are derived.

## A.1  Notation

Let $X$ be a Polish space and $K : X \times X \to \mathbb{R}$ a positive semidefinite function on $X$, we denote $\mathcal{H}$ the Hilbert space obtained by the completion of

$$\mathcal{H} = \overline{\mathrm{span}\{K(x, \cdot) \mid x \in X\}}$$

according to the norm induced by the inner product $\langle K(x, \cdot), K(x', \cdot)\rangle_{\mathcal{H}} = K(x, x')$. Spaces $\mathcal{H}$ constructed in this way are known as *reproducing kernel Hilbert spaces* and there is a one-to-one relation between a kernel $K$ and its associated RKHS. For more details on RKHS we refer the reader to [29, 30]. Given a kernel $K$, in the following we will denote with $K_x = K(x, \cdot) \in \mathcal{H}$ for all $x \in X$. We say that a kernel is bounded if $\|K_x\|_{\mathcal{H}} \leq \kappa$ with $\kappa > 0$. In the following we will always assume $K$ to be continuous and bounded by $\kappa > 0$. The continuity of $K$ with the fact that $X$ is Polish implies $\mathcal{H}$ to be separable [30].

In the rest of the appdendizes we denote with $A_\lambda$, the operator $A + \lambda I$, for any symmetric linear operator $A$, $\lambda \in \mathbb{R}$ and $I$ the identity operator.

## A.2  Definitions

For $n \in \mathbb{N}$, $(x_i)_{i=1}^n$, and $J \subseteq \{1, \ldots, n\}$, $A \in \mathbb{R}^{|J| \times |J|}$ diagonal matrix with positive diagonal, denote $\widetilde{\ell}_J$ in eq. (3) by showing the dependence from both $J$ and $A$ as

$$\widetilde{\ell}_{J,A}(i, \lambda) = (\lambda n)^{-1}(\widehat{K}_{ii} - \widehat{K}_{J,i}^\top (\widehat{K}_{J,J} + \lambda n A)^{-1} \widehat{K}_{J,i}). \tag{19}$$

Moreover define $\widehat{C}_{J,A}$ as follows

$$\widehat{C}_{J,A} = \frac{1}{|J|} \sum_{i=1}^{|J|} A_{ii}^{-1} K_{x_{j_i}} \otimes K_{x_{j_i}}.$$

We define the *out-of-sample leverage scores*, that are an extension of $\widetilde{\ell}_{J,A}$ to any point $x$ in the space $X$.

**Definition 1** (out-of-sample leverage scores). *Let $J = \{j_1, \ldots, j_M\} \subseteq \{1, \ldots, n\}$, with $M \in \mathbb{N}$ and $A \in \mathbb{R}^{M \times M}$ be a positive diagonal matrix. Then for any $x \in X$ and $\lambda > 0$ we define*

$$\widehat{\ell}_{J,A}(x, \lambda) = \frac{1}{n}\|(\widehat{C}_{J,A} + \lambda I)^{-1/2} K_x\|_{\mathcal{H}}^2.$$

*Moreover define $\widehat{\ell}_{\emptyset,[]}(x, \lambda) = (\lambda n)^{-1} K(x, x)$.*

In particular we denote by

$$\widehat{\ell}(x, \lambda) = \widehat{\ell}_{[n],I}(x, \lambda),$$

the out of sample version of the leverage scores $\ell(i, \lambda)$. Indeed note that $\widehat{\ell}(x_i, \lambda) = \ell(i, \lambda)$ for $i \in [n]$ and $\lambda > 0$ as proven by the next proposition that shows, more generally, the relation between $\widehat{\ell}_{J,A}$ and $\widetilde{\ell}_{J,A}$.

**Proposition 1.** *Let $n \in \mathbb{N}$, $(x_i)_{i=1}^n \subseteq X$. For any $\lambda > 0, J \subseteq \{1, \ldots, n\}, A \in \mathbb{R}^{|J| \times |J|}$ with $A$ positive diagonal, we that that for any $x \in X$, $\widehat{\ell}_{J,A}(x, \lambda)$ in Def. 1 and $\widetilde{\ell}_{J,A}(x, \lambda)$ in Def. 3, satisfy*

$$\widehat{\ell}_{J, \frac{n}{|J|}A}(x_i, \lambda) = \widetilde{\ell}_{J,A}(i, \lambda),$$

*when $|J| > 0$, and $\widehat{\ell}_{\emptyset,[]}(x_i, \lambda) = \widetilde{\ell}_{\emptyset,[]}(i, \lambda)$, when $|J| = 0$, for any $i \in [n], \lambda > 0$.*

*Proof.* Let $J = \{j_1, \ldots, j_{|J|}\}$. We will first show that $\widehat{\ell}_{J,A}(x, \lambda)$ is characterized by,

$$\widehat{\ell}_{J,A}(x, \lambda) = \frac{1}{\lambda n} K(x, x) - \frac{1}{\lambda n} v_J(x)^\top (K_J + \lambda |J| A)^{-1} v_J(x),$$

with $K_J \in \mathbb{R}^{M \times M}$ with $(K_J)_{lm} = K(x_{j_l}, x_{j_m})$ and $v_J(x) = (K(x, x_{j_1}), \ldots, K(x, x_{j_M}))$. Denote with $Z_J : \mathcal{H} \to \mathbb{R}^{|J|}$, the linear operator defined by $Z_J = (K_{x_{j_1}}, \ldots, K_{x_{j_{|J|}}})^\top$, that is $(Z_J f)_k = \langle K_{x_{j_k}}, f \rangle_{\mathcal{H}}$, for $f \in \mathcal{H}$ and $k \in \{1, \ldots |J|\}$. Then, by denoting with $B = |J| A$ we have

$$Z_J^* B^{-1} Z_J = \frac{1}{|J|} \sum_{i=1}^{|J|} A_{ii}^{-1} K_{x_{j_i}} \otimes K_{x_{j_i}} = \widehat{C}_{J,A}.$$

Now note that, since $(Q + \lambda I)^{-1} = \lambda^{-1}(I - Q(Q + \lambda I)^{-1})$ for any positive linear operator and $\lambda > 0$, we have

$$\widehat{\ell}_{J,A}(x, \lambda) = \frac{1}{n} \left\langle K_x, (\widehat{C}_{J,A} + \lambda I)^{-1} K_x \right\rangle_{\mathcal{H}} = \frac{1}{\lambda n} \left\langle K_x, (I - \widehat{C}_{J,A}(\widehat{C}_{J,A} + \lambda I)^{-1}) K_x \right\rangle_{\mathcal{H}}$$

$$= \frac{K(x, x)}{\lambda n} - \frac{1}{\lambda n} \left\langle K_x, Z_J^* B^{-1/2} (B^{-1/2} Z_J Z_J^* B^{-1/2} + \lambda I)^{-1} B^{-1/2} Z_J K_x \right\rangle_{\mathcal{H}},$$

where in the last step we use the fact that $R^* R(R^* R + \lambda I)^{-1} = R^* (RR^* + \lambda I)^{-1} R$, for any bounded linear operator $R$ and $\lambda > 0$. In particular we used it with $R = B^{-1/2} Z_J$. Now note that $Z_J Z_J^* \in \mathbb{R}^{|J| \times |J|}$ and in particular $Z_J Z_J^* = K_J$, moreover $Z_J K_x = v(x)$, so

$$\widehat{\ell}_{J,A}(x, \lambda) = \frac{K(x, x)}{\lambda n} - \frac{1}{\lambda n} v(x)^\top B^{-1/2} (B^{-1/2} K_J B^{-1/2} + \lambda I)^{-1} B^{-1/2} v(x)$$

$$= \frac{K(x, x)}{\lambda n} - \frac{1}{\lambda n} v(x)^\top (K_J + \lambda B)^{-1} v(x)$$

$$= \frac{K(x, x)}{\lambda n} - \frac{1}{\lambda n} v(x)^\top (K_J + \lambda |J| A)^{-1} v(x),$$

where in the second step we used the fact that $B^{-1/2}(B^{-1/2} K B^{-1/2} + \lambda I)^{-1} B^{-1/2} = (K + \lambda B)^{-1}$, for any invertible $B$ any positive operator $K$ and $\lambda > 0$.

Finally note that

$$\widehat{\ell}_{J, \frac{n}{|J|} A}(x_i, \lambda) = \frac{K(x, x)}{\lambda n} - \frac{1}{\lambda n} v(x)^\top (K_J + \lambda n A)^{-1} v(x) = \widetilde{\ell}_{J,A}(i, \lambda).$$

$\square$

## A.3 Preliminary results

Denote with $G_\lambda(A, B)$ the quantity

$$G_\lambda(A, B) = \|(A + \lambda I)^{-1/2}(A - B)(A + \lambda I)^{-1/2}\|,$$

for $A, B$ positive bounded linear operators and for $\lambda > 0$.

**Proposition 2.** *Let $A, B$ be positive bounded linear operators and $\lambda > 0$, then*

$$\|I - (A + \lambda I)^{-1/2}(B + \lambda I)(A + \lambda I)^{-1/2}\| = G_\lambda(A, B) \leq \frac{G_\lambda(B, A)}{1 - G_\lambda(B, A)},$$

*where the last inequality holds if $G_\lambda(B, A) < 1$.*

*Proof.* For the sake of compactness denote with $A_\lambda$ the operator $A + \lambda I$ and with $B_\lambda$ the operator $B + \lambda I$. First of all note that $I = A_\lambda^{-1/2} A_\lambda A_\lambda^{-1/2}$, so

$$I - A_\lambda^{-1/2} B_\lambda A_\lambda^{-1/2} = A_\lambda^{-1/2} A_\lambda A_\lambda^{-1/2} - A_\lambda^{-1/2} B_\lambda A_\lambda^{-1/2}$$

$$= A_\lambda^{-1/2}(A_\lambda - B_\lambda) A_\lambda^{-1/2} = A_\lambda^{-1/2}(A - B) A_\lambda^{-1/2}$$

$$= A_\lambda^{-1/2} B_\lambda^{1/2} \; B_\lambda^{-1/2}(A - B) B_\lambda^{-1/2} \; B_\lambda^{1/2} A_\lambda^{-1/2},$$

where in the last step we multiplied and divided by $B_\lambda^{1/2}$. Then

$$\|I - A_\lambda^{-1/2}B_\lambda A_\lambda^{-1/2}\| \le \|A_\lambda^{-1/2}B_\lambda^{1/2}\|^2\|B_\lambda^{-1/2}(A-B)B_\lambda^{-1/2}\|,$$

moreover, by Prop. 7 of [15] (see also Prop. 8 of [23]), if $G_\lambda(B,A) < 1$, we have

$$\|A_\lambda^{-1/2}B_\lambda^{1/2}\|^2 \le (1 - G_\lambda(B,A))^{-1}.$$

$\square$

**Proposition 3.** *Let $A, B, C$ be bounded positive linear operators on a Hilbert space. Let $\lambda > 0$. Then, the following holds*

$$G_\lambda(A,C) \le G_\lambda(A,B) + (1 + G_\lambda(A,B))G_\lambda(B,C).$$

*Proof.* In the following we denote with $A_\lambda$ the operator $A + \lambda I$ and the same for $B, C$. Then

$$\|A_\lambda^{-1/2}(A-C)A_\lambda^{-1/2}\| \le \|A_\lambda^{-1/2}(A-B)A_\lambda^{-1/2}\| + \|A_\lambda^{-1/2}(B-C)A_\lambda^{-1/2}\|.$$

Now note that, by dividing and multiplying for $B_\lambda^{1/2}$, we have

$$\|A_\lambda^{-1/2}(B-C)A_\lambda^{-1/2}\| = \|A_\lambda^{-1/2}B_\lambda^{1/2} \ B_\lambda^{-1/2}(B-C)B_\lambda^{-1/2}B_\lambda^{1/2}A_\lambda^{-1/2}\|$$
$$\le \|A_\lambda^{-1/2}B_\lambda^{1/2}\|^2\|B_\lambda^{-1/2}(B-C)B_\lambda^{-1/2}\| = \|A_\lambda^{-1/2}B_\lambda^{1/2}\|^2 G_\lambda(B,C).$$

Finally note that, since $\|Z\|^2 = \|Z^*Z\|$ for any bounded linear operator $Z$, we have

$$\|A_\lambda^{-1/2}B_\lambda^{1/2}\|^2 = \|A_\lambda^{-1/2}B_\lambda A_\lambda^{-1/2}\| = \|I + (I - A_\lambda^{-1/2}B_\lambda A_\lambda^{-1/2})\| \le 1 + \|I - A_\lambda^{-1/2}B_\lambda A_\lambda^{-1/2}\|.$$

Moreover, by Prop. 2, we have that

$$\|I - A_\lambda^{-1/2}B_\lambda A_\lambda^{-1/2}\| = G_\lambda(A,B).$$

$\square$

**Proposition 4.** *Let $B$ be a bounded linear operator, then*

$$1 - \|I - BB^*\| \le \sigma_{\min}(B)^2 \le \sigma_{\max}(B)^2 \le 1 + \|I - BB^*\|.$$

*Proof.* Now we recall that, denoting by $\preceq$ the Lowner partial order, for a positive bounded operator $A$ such that $aI \preceq A \preceq bI$ for $0 \le a \le b$, we have $(1-b)I \preceq I - A \preceq (1-a)I \preceq (1+b)I$ and so, since $BB^* = I - (I - BB^*)$, we have

$$(1 - \|I - BB^*\|)I \preceq \sigma_{\min}(B)^2 I \preceq BB^* \preceq \sigma_{\max}(B)^2 I \preceq 1 + (1 + \|I - BB^*\|)I,$$

from we have the desired result. $\square$

Let $\|\cdot\|_{HS}$ denote the Hilbert-Schmidt norm.

We recall and adapt to our needs a result from Prop. 8 of [15].

**Proposition 5.** *Let $\lambda > 0$ and $v_1, \dots, v_n$ with $n \ge 1$, be identically distributed random vectors on separable Hilbert space $\mathcal{H}$, such that there exists $\kappa^2 > 0$ for which $\|v\|_{\mathcal{H}} \le \kappa^2$ almost surely. Denote by $Q$ the Hermitian operator $Q = \frac{1}{n}\sum_{i=1}^n \mathbb{E}[v_i \otimes v_i]$. Let $Q_n = \frac{1}{n}\sum_{i=1}^n v_i \otimes v_i$. Then for any $\delta \in (0,1]$, the following holds*

$$\|(Q + \lambda I)^{-1/2}(Q - Q_n)(Q + \lambda I)^{-1/2}\| \le \frac{4\kappa^2\beta}{3\lambda n} + \sqrt{\frac{2\kappa^2\beta}{\lambda n}}$$

*with probability $1 - \delta$ and $\beta = \log \frac{4\operatorname{Tr}(Q(Q+\lambda I)^{-1})}{\|Q(Q+\lambda I)^{-1}\|\delta} \le \frac{8\kappa^2(1 + \operatorname{Tr}(Q_\lambda^{-1}Q))}{\|Q\|\delta}$.*

*Proof.* Let $Q_\lambda = Q + \lambda I$. Here we apply non-commutative Bernstein inequality like [3] (with the extension to separable Hilbert spaces as in[15], Prop. 12) on the random variables $Z_i = M - Q_\lambda^{-1/2} v_i \otimes Q_\lambda^{-1/2} v_i$ with $M_i = Q_\lambda^{-1/2}(\mathbb{E}[v_i \otimes v_i])Q_\lambda^{-1/2}$ for $1 \le i \le n$. Note that the expectation of $Z_i$ is 0. The random vectors are bounded by

$$\|Q_\lambda^{-1/2} v_i \otimes Q_\lambda^{-1/2} v_i - M_i\| = \|\mathbb{E}_{v_i'}[Q_\lambda^{-1/2} v_i' \otimes Q_\lambda^{-1/2} v_i' - Q_\lambda^{-1/2} v_i \otimes Q_\lambda^{-1/2} v_i]\|_{\mathcal{H}}$$

$$\le 2\|\kappa^2\|\|(Q + \lambda)^{-1/2}\|^2 \le \frac{2\kappa^2}{\lambda},$$

and the second orded moment is

$$\mathbb{E}(Z_i)^2 = \mathbb{E} \ \langle v_i, Q_\lambda^{-1} v_i \rangle \ Q_\lambda^{-1/2} v_i \otimes Q_\lambda^{-1/2} v_i \quad - \quad Q_\lambda^{-2} Q^2$$

$$\le \frac{\kappa^2}{\lambda}\mathbb{E}[Q_\lambda^{-1/2} v_1 \otimes Q_\lambda^{-1/2} v_1] = \frac{\kappa^2}{\lambda}Q(Q + \lambda I)^{-1} =: S.$$

Now we can apply the Bernstein inequality with *intrinsic dimension* in [3] (or Prop. 12 in [15]). Now some considerations on $\beta$. It is $\beta = \log \frac{4\operatorname{Tr} S}{\|S\|\delta} = \frac{4\operatorname{Tr} Q_\lambda^{-1} Q}{\|Q_\lambda^{-1} Q\|\delta}$, now we need a lower bound for $\|Q_\lambda^{-1} Q\| = \frac{\sigma_1}{\sigma_1 + \lambda}$ where $\sigma_1 = \|Q\|$ is the biggest eigenvalue of $Q$, now, when $0 < \lambda \le \sigma_1$ we have $\beta \le \frac{8\operatorname{Tr} Q}{\lambda\delta}$.

When $\lambda \ge \sigma_1$, note that $\operatorname{Tr}(Q(Q + \lambda I)^{-1}) \le \lambda^{-1}\operatorname{Tr}(Q) \le \kappa^2/\lambda$, then

$$\frac{\operatorname{Tr}(Q(Q + \lambda I)^{-1})}{\|Q_\lambda^{-1} Q\|} \le \frac{\kappa^2}{\lambda \frac{\sigma_1}{\sigma_1 + \lambda}} = \frac{\kappa^2}{\lambda} + \frac{\kappa^2}{\sigma_1} \le \frac{2\kappa^2}{\sigma_1}.$$

So finally $\beta \le \frac{8(\kappa^2/\|Q\| + \operatorname{Tr}(Q_\lambda^{-1} Q))}{\delta}$ $\qquad\qquad\qquad\qquad\qquad\qquad\qquad\qquad\qquad\square$

### A.4 Analytic decomposition

**Lemma 2.** *Let* $\lambda > 0$, $J, J' \subseteq \{1, \dots, n\}$, *with* $|J|, |J'| \ge 1$ *and* $A \in \mathbb{R}^{|J| \times |J|}$, $A' \in \mathbb{R}^{|J'| \times |J'|}$ *positive diagonal matrices, then*

$$\frac{1 - 2\nu}{1 - \nu}\widehat{\ell}_{J',A'}(x, \lambda) \le \widehat{\ell}_{J,A}(x, \lambda) \le \frac{1}{1 - \nu}\widehat{\ell}_{J',A'}(x, \lambda), \quad \forall x \in X,$$

*with* $\nu = G_\lambda(\widehat{C}_{J',A'}, \widehat{C}_{J,A})$.

*Proof.* By denoting with $B$ the operator

$$B = (\widehat{C}_{J,A} + \lambda I)^{-1/2}(\widehat{C}_{J',A'} + \lambda I)^{1/2},$$

and according to the characterization of $\widehat{\ell}_{J,A}(x, \lambda)$ via Prop. 1, we have

$$\widehat{\ell}_{J,A}(x, \lambda) = n^{-1}\|(\widehat{C}_{J,A} + \lambda I)^{-1/2} K_x\|_{\mathcal{H}}^2 = n^{-1}\|B \ (\widehat{C}_{J',A'} + \lambda I)^{-1/2} K_x\|_{\mathcal{H}}^2.$$

So, by recalling the fact that, by definition of Lowner partial order $\preceq$, we have $a\|v\|^2 \le \|Av\|^2 \le b\|v\|^2$, for any vector $v$ and bounded linear operator such that $aI \preceq A^*A \preceq bI$ with $0 \le a \le b$, and the fact that $\sigma(A^*A) = \sigma(AA^*) = \sigma(A)^2$, we have

$$\sigma_{\min}(B)^2\|(\widehat{C}_{J',A'} + \lambda I)^{-1/2} K_x\|_{\mathcal{H}}^2 \le \|B(\widehat{C}_{J',A'} + \lambda I)^{-1/2} K_x\|_{\mathcal{H}}^2 \le \sigma_{\max}(B)^2\|(\widehat{C}_{J',A'} + \lambda I)^{-1/2} K_x\|_{\mathcal{H}}^2.$$

That, by Prop. 1, is equivalent to

$$\sigma_{\min}(B)^2\widehat{\ell}_{J',A'}(x, \lambda) \le \widehat{\ell}_{J,A}(x, \lambda) \le \sigma_{\max}(B)^2\widehat{\ell}_{J',A'}(x, \lambda).$$

By Prop. 4 we have $1 - \|I - BB^*\| \le \sigma_{\min}(B)^2 \le \sigma_{\max}(B)^2 \le 1 + \|I - BB^*\|$. Finally, by Prop. 2, we have

$$\|I - BB^*\| \le \frac{\nu}{1 - \nu}.$$

$\qquad\qquad\qquad\qquad\qquad\qquad\qquad\qquad\qquad\qquad\qquad\qquad\qquad\qquad\qquad\square$

**Lemma 3.** *Let $0 < \lambda \le \lambda'$, and $J \subseteq \{1, \dots, n\}$ and $A \in \mathbb{R}^{|J| \times |J|}$, then*

$$\widehat{\ell}_{J,A}(x, \lambda') \le \widehat{\ell}_{J,A}(x, \lambda) \le \frac{\lambda'}{\lambda} \widehat{\ell}_{J,A}(x, \lambda'), \quad \forall x \in X.$$

*Proof.* If $|J| = 0$ we have that $\widehat{\ell}_{\emptyset, []}(x, \lambda) = \frac{K(x,x)}{\lambda n}$ and the desired result is easily verified. If $|J| \ge 1$, let $B = (C_{J,A} + \lambda I)^{-1/2}(C_{J,A} + \lambda' I)^{1/2}$. By recalling the fact that, by definition of Lowner partial order $\preceq$, we have $a\|v\|^2 \le \|Av\|^2 \le b\|v\|^2$, for any vector $v$ and bounded linear operator such that $aI \preceq A^*A \preceq bI$ with $0 \le a \le b$, and the fact that $\sigma(A^*A) = \sigma(AA^*) = \sigma(A)^2$, we have

$$\sigma_{\min}(B)^2 \|(\widehat{C}_{J,A} + \lambda' I)^{-1/2} K_x\|_{\mathcal{H}}^2 \le \|B(\widehat{C}_{J,A} + \lambda' I)^{-1/2} K_x\|_{\mathcal{H}}^2 \le \sigma_{\max}(B)^2 \|(\widehat{C}_{J,A} + \lambda' I)^{-1/2} K_x\|_{\mathcal{H}}^2.$$

That, by Prop. 1, is equivalent to

$$\sigma_{\min}(B)^2 \widehat{\ell}_{J,A}(x, \lambda') \le \widehat{\ell}_{J,A}(x, \lambda) \le \sigma_{\max}(B)^2 \widehat{\ell}_{J,A}(x, \lambda').$$

Now note that

$$\sigma_{\min}(B)^2 \ge \inf_{\sigma \ge 0} \frac{\sigma + \lambda'}{\sigma + \lambda} = 1, \quad \sigma_{\max}(B)^2 \ge \sup_{\sigma \ge 0} \frac{\sigma + \lambda'}{\sigma + \lambda} = \frac{\lambda'}{\lambda}.$$

$\square$

**Theorem 3.** *Let $\lambda > 0$, $J \subseteq \{1, \dots, n\}$, with $|J| \ge 1$ and $A \in \mathbb{R}^{|J| \times |J|}$ positive diagonal. Then the following hold for any $x \in X$,*

$$\frac{1 - 2\nu_{J,A}}{1 - \nu_{J,A}} \widehat{\ell}(x, \lambda) \le \widehat{\ell}_{J,A}(x, \lambda) \le \frac{1}{1 - \nu_{J,A}} \widehat{\ell}(x, \lambda),$$

*where $\nu_{J,A} = G_\lambda(\widehat{C}, \widehat{C}_{J,A})$. Morever note that for any $|U| \subseteq \{1, \dots, n\}$, we have*

$$\nu_{J,A} \le \eta_U + (1 + \eta_U)\beta_{J,A,U},$$

*with $\beta_{J,A,U} = G_\lambda(\widehat{C}_{U,I}, \widehat{C}_{J,A})$ and $\eta_U = G_\lambda(\widehat{C}, \widehat{C}_{U,I})$.*

*Proof.* By applying Lemma 2, with their $J' = \{1, \dots, n\}$, $A' = I$, and recalling that $\widehat{\ell}(x, \lambda) = \widehat{\ell}_{\{1, \dots, n\}, I}$, we have for all $x \in X$

$$\frac{1 - 2\nu_{J,A}}{1 - \nu_{J,A}} \widehat{\ell}(x, \lambda) \le \widehat{\ell}_{J,A}(x, \lambda) \le \frac{1}{1 - \nu_{J,A}} \widehat{\ell}(x, \lambda).$$

To conclude the proof we bound $\nu_{J,A}$ in terms of $\beta_{J,A,U}$ and $\eta_U$, via Prop. 3. $\square$

### A.5 Proof for Algorithm 1

**Lemma 4.** *Let $n \in \mathbb{N}$, $(x_i)_{i=1}^n \subseteq X$. Let $U \subseteq \{1, \dots n\}$, with $|U| \ge 1$. Let $(p_k)_{k=1}^{|U|} \subset \mathbb{R}$ be a non-negative sequence summing to $1$. Let $M \in \mathbb{N}$ and $J = \{j_1, \dots, j_M\}$ with $j_i$ sampled i.i.d. from $\{1, \dots, |U|\}$ with probability $(p_k)_{k=1}^{|U|}$ and $A = |U| diag(p_{j_1}, \dots, p_{j_M})$. Let $\tau \in (0, 1]$, and $s := \sup_{k \in \{1, \dots, |U|\}} \frac{1}{|U|p_k} \|(\widehat{C}_{U,I} + \lambda I)^{-1/2} K_{x_{u_k}}\|_{\mathcal{H}}^2$. When*

$$M \ge 2s \log \frac{4n}{\tau},$$

*then the following holds with probability at least $1 - \tau$*

$$\|(\widehat{C}_{U,I} + \lambda I)^{-1/2}(\widehat{C}_{J,A} - \widehat{C}_{U,I})(\widehat{C}_{U,I} + \lambda I)^{-1/2}\| \le \sqrt{\frac{4s \log \frac{4n}{\tau}}{M}}.$$

*Proof.* Denote with $\zeta_i$ the random variable

$$\zeta_i = \frac{1}{|U|p_k}(\widehat{C}_{U,I} + \lambda I)^{-1/2}(K_{x_{j_i}} \otimes K_{x_{j_i}})(\widehat{C}_{U,I} + \lambda I)^{-1/2},$$

for $i \in \{1, \ldots, M\}$. In particular note that $\zeta_1, \ldots, \zeta_M$ are i.i.d. since $j_1, \ldots, j_M$ are. Moreover note the following two facts

$$\|\zeta_i\| = \sup_{k \in \{1, \ldots, |U|\}} \frac{1}{|U|p_k} \|(\widehat{C}_{U,I} + \lambda I)^{-1/2} K_{x_{u_k}}\|_{\mathcal{H}}^2 = s,$$

$$\mathbb{E}[\zeta_i] = \sum_{k=1}^{|U|} p_k \frac{1}{|U|p_k} (\widehat{C}_{U,I} + \lambda I)^{-1/2} (K_{x_k} \otimes K_{x_k})(\widehat{C}_{U,I} + \lambda I)^{-1/2}$$

$$= (\widehat{C}_{U,I} + \lambda I)^{-1/2} \widehat{C}_{U,I} (\widehat{C}_{U,I} + \lambda I)^{-1/2} =: W,$$

where for the second identity we used the fact that $d/l_k = 1/(p_k|U|)$. Since by definition of $\widehat{C}_{J,A}$ we have

$$\frac{1}{M} \sum_{i=1}^{M} \zeta_i = (\widehat{C}_{U,I} + \lambda I)^{-1/2} \left( \frac{1}{|J|} \sum_{i=1}^{M} \frac{1}{A_{ii}} K_{x_{j_i}} \otimes K_{x_{j_i}} \right) (\widehat{C}_{U,I} + \lambda I)^{-1/2}$$

$$= (\widehat{C}_{U,I} + \lambda I)^{-1/2} \widehat{C}_{J,A} (\widehat{C}_{U,I} + \lambda I)^{-1/2},$$

then, by applying non-commutative Bernstein inequality (Prop. 5 is a version specific for our problem), we have

$$\|(\widehat{C}_{U,I} + \lambda I)^{-1/2}(\widehat{C}_{J,A} - \widehat{C}_{U,I})(\widehat{C}_{U,I} + \lambda I)^{-1/2}\| = \Big\| \frac{1}{M} \sum_{i=1}^{M} (\zeta_i - \mathbb{E}[\zeta_i]) \Big\| \le \frac{2s\eta}{3M} + \sqrt{\frac{2s\|W\|\eta}{M}},$$

with probability at least $1 - \tau$, and $\eta := \log \frac{4 \operatorname{Tr}(W)}{\tau \|W\|}$. In particular, by noting that $\|W\| \le 1$ by definition, when $M \ge 2s\eta$, then

$$\frac{2s\eta}{3M} + \sqrt{\frac{2s\|W\|\eta}{M}} \le \frac{2s\eta}{3M} + \sqrt{\frac{2s\eta}{M}} \le \frac{1}{3}\sqrt{\frac{2s\eta}{M}} + \sqrt{\frac{2s\eta}{M}} \le \sqrt{\frac{4s\eta}{M}}.$$

To conclude note that $\frac{\operatorname{Tr}(W)}{\|W\|} \le \operatorname{rank}(W) \le |U| \le n$, so $\eta \le \log \frac{4n}{\tau}$. $\qquad \square$

**Lemma 5.** *Let* $n, R \in \mathbb{N}$, $(x_i)_{i=1}^{n} \subseteq X$. *Let* $U = \{u_1, \ldots, u_R\}$ *with* $u_i$ *i.i.d. with uniform probability on* $\{1, \ldots, n\}$. *Let* $\tau \in (0, 1]$ *and let* $\lambda > 0$. *When*

$$R \ge \frac{2n\kappa^2}{\lambda n + \kappa^2} \log \frac{4n}{\tau},$$

*then the following holds with probability* $1 - \tau$

$$\|(\widehat{C} + \lambda I)^{-1/2}(\widehat{C}_{U,I} - \widehat{C})(\widehat{C} + \lambda I)^{-1/2}\| \le \sqrt{\frac{4n\kappa^2 \log \frac{4n}{\tau}}{(\lambda n + \kappa^2)R}}.$$

*Proof.* Denote by $\zeta_i$ the random variable $\zeta_i = (\widehat{C} + \lambda I)^{-1/2}(K_{x_{u_i}} \otimes K_{x_{u_i}})(\widehat{C} + \lambda I)^{-1/2}$, for $i \in \{1, \ldots, R\}$. Note that $\zeta_i$ are i.i.d. since $u_i$ are. Moreover note that

$$\|\zeta_i\| = \sup_{i \in \{1, \ldots, n\}} \|(\widehat{C} + \lambda I)^{-1/2} K_{x_i}\|^2 \le \sup_{i \in \{1, \ldots, n\}} \|(\frac{1}{n} K_{x_i} \otimes K_{x_i} + \lambda I)^{-1/2} K_{x_i}\|^2$$

$$\le \frac{n\kappa^2}{\lambda n + \kappa^2} =: v.$$

Moreover note that

$$\mathbb{E}[\zeta_i] = \frac{1}{n} \sum_{i=1}^{n} (\widehat{C} + \lambda I)^{-1/2}(K_{x_i} \otimes K_{x_i})(\widehat{C} + \lambda I)^{-1/2} = (\widehat{C} + \lambda I)^{-1/2} \widehat{C} (\widehat{C} + \lambda I)^{-1/2} =: W.$$

So we have, by non-commutative Bernstein inequality (Prop. 5 is a version specific for our problem),

$$\|(\widehat{C} + \lambda I)^{-1/2}(\widehat{C}_{U,I} - \widehat{C})(\widehat{C} + \lambda I)^{-1/2}\| = \Big\| \frac{1}{M} \sum_{i=1}^{M} (\zeta_i - \mathbb{E}[\zeta_i]) \Big\| \le \frac{2v\eta}{3R} + \sqrt{\frac{2v\|W\|\eta}{R}},$$

with probability at least $1 - \tau$, and $\eta := \log \frac{4\operatorname{Tr}(W)}{\tau \|W\|}$. In particular, by noting that $\|W\| \leq 1$ by definition, when $R \geq \frac{2n\kappa^2 \eta}{(\lambda n + \kappa^2)R}$, analogously to the end of the proof of Lemma 4, we have $\frac{2v\eta}{3R} + \sqrt{\frac{2v\|W\|\eta}{R}} \leq \sqrt{\frac{4n\kappa^2 \eta}{(\lambda n + \kappa^2)R}}$. To conclude note that $\frac{\operatorname{Tr}(W)}{\|W\|} \leq \operatorname{rank}(W) \leq n$, so $\eta \leq \log \frac{4n}{\tau}$. $\square$

**Lemma 6.** *Let* $n, R \in \mathbb{N}$, $(x_i)_{i=1}^n \subseteq X$. *Let* $U = \{u_1, \ldots, u_R\}$ *with* $u_i$ *i.i.d. with uniform probability on* $\{1, \ldots, n\}$. *Let* $\tau \in (0, 1]$ *and let* $\lambda > 0$. *When*

$$R \geq \frac{16n\kappa^2}{\lambda n + \kappa^2} \log \frac{4n}{\tau},$$

*then the following holds with probability* $1 - \tau$

$$\frac{n}{R} \sum_{i=1}^R \widehat{\ell}(x_{u_i}, \lambda) < \max\left(5, \frac{6}{5} d_{\mathit{eff}}(\lambda)\right).$$

*Proof.* First of all denote with $z_i$ the random variable $z_i = \frac{n}{R}\widehat{\ell}(x_{u_i}, \lambda)$ and note that $(z_i)_{i=1}^R$ are i.i.d. since $(u_i)_{i=1}^R$ are. Moreover, by the characterization of $\widehat{\ell}(x, \lambda)$ via Prop. 1, we have

$$|z_i| \leq \sup_{k \in \{1, \ldots, n\}} \|(\widehat{C} + \lambda I)^{-1/2} K_{x_k}\|^2 \leq \|(K_{x_k} \otimes K_{x_k}/n + \lambda I)^{1/2} K_{x_k}\|^2 \leq \frac{\kappa^2}{R(\kappa^2/n + \lambda)} =: v,$$

moreover we have

$$\mathbb{E}[z_i] = \mathbb{E}[\operatorname{Tr}((\widehat{C} + \lambda I)^{-1}(K_{x_{u_i}} \otimes K_{x_{u_i}}))] = \operatorname{Tr}((\widehat{C} + \lambda I)^{-1}\mathbb{E}[K_{x_{u_i}} \otimes K_{x_{u_i}}])$$

$$= \operatorname{Tr}\left((\widehat{C} + \lambda I)^{-1} \sum_{k=1}^n \frac{1}{n} K_{x_k} \otimes K_{x_k}\right) = \operatorname{Tr}\left((\widehat{C} + \lambda I)^{-1}\widehat{C}\right) = d_{\mathrm{eff}}(\lambda).$$

So by applying Bernstein inequality, the following holds with probability at least $1 - \tau$

$$\left|\frac{n}{R} \sum_{i=1}^R \widehat{\ell}(x_{u_i}, \lambda) - d_{\mathrm{eff}}(\lambda)\right| = \left|\frac{1}{R} \sum_{i=1}^R (z_i - \mathbb{E}[z_i])\right| \leq \frac{2v \log \frac{2}{\tau}}{3R} + \sqrt{\frac{2v d_{\mathrm{eff}}(\lambda) \log \frac{2}{\tau}}{3R}}.$$

So we have

$$\frac{n}{R} \sum_{i=1}^R \widehat{\ell}(x_{u_i}, \lambda) \leq d_{\mathrm{eff}}(\lambda) + \left|\frac{n}{R} \sum_{i=1}^R \widehat{\ell}(x_{u_i}, \lambda) - d_{\mathrm{eff}}(\lambda)\right| \leq d_{\mathrm{eff}}(\lambda) + \frac{2v \log \frac{2}{\tau}}{3R} + \sqrt{\frac{2v d_{\mathrm{eff}}(\lambda) \log \frac{2}{\tau}}{R}}.$$

Now, if $d_{\mathrm{eff}}(\lambda) \leq 4$, since $R \geq 16v \log \frac{2}{\tau}$, we have that

$$d_{\mathrm{eff}}(\lambda) + \frac{2v \log \frac{2}{\tau}}{3R} + \sqrt{\frac{2v d_{\mathrm{eff}}(\lambda) \log \frac{2}{\tau}}{R}} \leq 4 + \frac{1}{24} + \sqrt{\frac{1}{2}} < 5.$$

If $d_{\mathrm{eff}}(\lambda) > 4$, since $R \geq 16v \log \frac{2}{\tau}$, we have

$$d_{\mathrm{eff}}(\lambda) + \frac{2v \log \frac{2}{\tau}}{3R} + \sqrt{\frac{2v d_{\mathrm{eff}}(\lambda) \log \frac{2}{\tau}}{3R}} \leq \left(1 + \frac{1}{24 d_{\mathrm{eff}}(\lambda)} + \sqrt{\frac{1}{8 d_{\mathrm{eff}}(\lambda)}}\right) d_{\mathrm{eff}}(\lambda) < \frac{6}{5} d_{\mathrm{eff}}(\lambda).$$

$\square$

**Theorem 4.** *Let* $n \in \mathbb{N}$, $(x_i)_{i=1}^n \subseteq X$. *Let* $\delta \in (0, 1]$, $t, q > 1$, $\lambda > 0$ *and* $H, d_h, \lambda_h, J_h, A_h, U_h$ *as in Alg. 1. Let* $\bar{A}_h = \frac{n}{|J|} A_h$ *and* $\nu_h = G_{\lambda_h}(\widehat{C}, \widehat{C}_{J_h, \bar{A}_h})$, $\beta_h = G_{\lambda_h}(\widehat{C}_{U_h, I}, \widehat{C}_{J_h, \bar{A}_h})$, $\eta_h = G_{\lambda_h}(\widehat{C}, \widehat{C}_{U_h, I})$. *When*

$$\lambda_0 = \frac{\kappa^2}{\min(t, 1)}, \quad q_1 \geq \frac{5\kappa^2 q_2}{q(1+t)}, \quad q_2 \geq 12q \frac{(2t+1)^2}{t^2}(1+t) \log \frac{12Hn}{\delta},$$

*then the following holds with probability $1 - \delta$: for any $h \in \{0, \dots, H\}$*

$$a) \qquad \frac{1}{T}\widehat{\ell}(x, \lambda_h) \leq \widehat{\ell}_{J_h, \bar{A}_h}(x) \leq \min(T, 2)\widehat{\ell}(x, \lambda_h), \quad \forall x \in X,$$

$$b) \qquad d_h \leq 3q \, d_{eff}(\lambda_h) \vee 10q, \quad and \quad |J_h| \leq q_2(3qd_{eff}(\lambda_h) \vee 10q). \qquad (20)$$

$$c) \qquad \beta_h \leq \frac{7}{11c_T}, \quad \eta_h \leq \frac{3}{11c_T}, \quad \nu_h \leq \frac{1}{c_T}.$$

*where $T = 1 + t$ and $c_T = 2 + 1/(T-1)$.*

*Proof.* Let $H$, $c_T$, $q$ and $\lambda_h, U_h, J_h, A_h, d_h, P_h = (p_{h,k})_{k=1}^{R_h}$, for $h \in \{0, \dots, H\}$ as defined in Alg. 1 and define $\tau = \delta/(3H)$. Now we are going to define some events and we prove a recurrence relation that they satisfy. Finally we unroll the recurrence relation and bound the resulting events in probability.

**Definitions of the events** Now we are going to define some events that will be useful to prove the theorem. Denote with $E_h$ the event such that the conditions in Eq. (20)-(a) hold for $J_h, A_h, U_h$. Denote with $F_h$ the event such that

$$\frac{n}{R_h} \sum_{u \in U_h} \widehat{\ell}(x_u, \lambda_{h-1}) \leq \frac{6}{5} d_{\text{eff}}(\lambda).$$

Denote with $B_{1,h}$ the event such that $\beta_h$, satisfies

$$\beta_h \leq \sqrt{\frac{4s_h \log \frac{4n}{\tau}}{M_h}}, \quad \text{with} \quad s_h := \sup_{k \in \{1, \dots, R_h\}} \frac{1}{R_h p_{h,k}} \|(\widehat{C}_{U_h, I} + \lambda_h I)^{-1/2} K_{x_{u_k}}\|^2. \qquad (21)$$

Denote with $B_{2,h}$ the event such that $\eta_h$, satisfies

$$\eta_h \leq \sqrt{\frac{4\kappa^2 n \log \frac{\kappa^2}{\lambda_h \tau}}{(\lambda_h n + \kappa^2)R_h}}.$$

**First bound for $s_h$.** Note that, by definition of $p_{h,k}$, that is, by Prop. 1

$$p_{h,k} = n\widetilde{\ell}_{J_{h-1}, A_{h-1}}(x_{u_k}, \lambda_h)/(d_h R_h) = n\widehat{\ell}_{J_{h-1}, \bar{A}_{h-1}}(x_{u_k}, \lambda_h)/(d_h R_h),$$

so

$$s_h = \sup_{k \in \{1, \dots, R_h\}} \frac{d_h \|(\widehat{C}_{U_h, I} + \lambda_h I)^{-1/2} K_{x_{u_k}}\|^2}{n\widehat{\ell}_{J_{h-1}, \bar{A}_{h-1}}(x_{u_k}, \lambda_h)} = \sup_{u \in U_h} \frac{d_h \widehat{\ell}_{U_h, I}(x_u, \lambda_h)}{\widehat{\ell}_{J_{h-1}, \bar{A}_{h-1}}(x_u, \lambda_h)},$$

where the last step consists in apply the definition of $\widehat{\ell}_{U_h, I}$. By applying Lemma 2 and 3 to $\widehat{\ell}_{U_h, I}(x, \lambda_h)$, we have

$$\widehat{\ell}_{U_h, I}(x, \lambda_h) \leq \frac{1}{1 - \eta_h}\widehat{\ell}(x, \lambda_h) \leq \frac{\lambda_{h-1}}{\lambda_h(1 - \eta_h)}\widehat{\ell}(x, \lambda_{h-1})$$

and analogously by applying Lemma 3 to $\widehat{\ell}_{J_{h-1}, \bar{A}_{h-1}}(x, \lambda_h)$, we have $\widehat{\ell}_{J_{h-1}, \bar{A}_{h-1}}(x, \lambda_h) \geq \widehat{\ell}_{J_{h-1}, \bar{A}_{h-1}}(x, \lambda_{h-1})$. So, by extending the sup of $s_h$ to the whole $X$, we have

$$s_h \leq d_h \sup_{x \in X} \frac{\widehat{\ell}_{U_h, I}(x, \lambda_h)}{\widehat{\ell}_{J_{h-1}, \bar{A}_{h-1}}(x, \lambda_h)} \leq \frac{\lambda_{h-1} d_h}{\lambda_h(1 - \eta_h)} \sup_{x \in X} \frac{\widehat{\ell}(x, \lambda_{h-1})}{\widehat{\ell}_{J_{h-1}, \bar{A}_{h-1}}(x, \lambda_{h-1})}.$$

Now we are ready to prove the recurrence relation, for $h \in \{1, \dots H\}$,

$$E_h \supseteq B_{1,h} \cap B_{2,h} \cap E_{h-1} \cap F_h.$$

**Analysis of $E_0$.** Note that, since $\|\widehat{C}\| \leq \kappa^2$, then $\frac{1}{\kappa^2 + \lambda}I \preceq (\widehat{C} + \lambda I)^{-1} \preceq \frac{1}{\lambda}$, so for any $x \in X$ the following holds

$$\frac{K(x,x)}{(\kappa^2 + \lambda)n} \leq \widehat{\ell}(x, \lambda) \leq \frac{K(x,x)}{\lambda n}.$$

Since $\lambda_0 = \frac{\kappa^2}{\min(2,T) - 1}$ and $\widehat{\ell}_{\emptyset,[]}(x, \lambda_0) = \frac{K(x,x)}{\lambda_0 n}$, we have

$$\frac{1}{T}\widehat{\ell}(x, \lambda_0) \leq \frac{1}{T}\frac{K(x,x)}{\lambda n} \leq \ell_{\emptyset,[]}(x, \lambda_0) = \frac{K(x,x)}{\lambda_0 n} = \frac{\min(2,T)K(x,x)}{(\kappa^2 + \lambda_0)n} \leq \min(2,T)\widehat{\ell}(x, \lambda_0).$$

Setting conventionally $d_0, \nu_0, \eta_0, \beta_0 = 0$ (they are not used by the algorithm or the proof), we have that $E_0$ holds everywhere and so, with probability 1.

**Analysis of $E_{h-1} \cap B_{1,h} \cap B_{2,h}$.** First note that under $E_{h-1}$, the following holds $\widehat{\ell}_{J_{h-1}, \bar{A}_{h-1}}(x, \lambda_{h-1}) \geq \frac{1}{T}\widehat{\ell}(x, \lambda_{h-1})$ and so

$$s_h \leq \frac{\lambda_{h-1}d_h}{\lambda_h(1 - \eta_h)} \sup_{x \in X} \frac{\widehat{\ell}(x, \lambda_{h-1})}{\widehat{\ell}_{J_{h-1}, \bar{A}_{h-1}}(x, \lambda_{h-1})} \leq \frac{\lambda_{h-1}d_h}{\lambda_h(1 - \eta_h)} \sup_{x \in X} \frac{\widehat{\ell}(x, \lambda_{h-1})}{\frac{1}{T}\widehat{\ell}(x, \lambda_{h-1})} \leq \frac{T\lambda_{h-1}d_h}{\lambda_h(1 - \eta_h)}.$$

Now note that under $B_{2,h}$, by applying the definition of $R_h$ in Alg. 1, by the condition on $q_1$, we have

$$\eta_h \leq \sqrt{\frac{4\kappa^2 n \log \frac{\kappa^2}{\lambda_h \tau}}{(\lambda_h n + \kappa^2)R_h}} \leq \sqrt{\frac{4 \log \frac{\kappa^2}{\lambda_h \tau}}{q_1}} \leq 3/(11c_T) \leq 3/22.$$

So under $B_{1,h} \cap B_{2,h} \cap E_{h-1}$ and the fact that $q = \frac{\lambda_{h-1}}{\lambda_h}$, we have $s_h \leq \frac{T\lambda_{h-1}d_h}{\lambda_h(1 - \eta_h)} \leq (8/7)qTd_h$ and so, since $M_h = q_2 d_h$, by the condition on $q_2$, we have

$$\beta_h \leq \sqrt{\frac{4s_h \log \frac{4n}{\tau}}{M_h}} \leq \sqrt{\frac{(32/7)qTd_h \log \frac{4n}{\tau}}{M_h}} = \sqrt{\frac{(32/7)qT \log \frac{4n}{\tau}}{q_2}} < \frac{7}{11c_T},$$

where in the last step we used the definition of $M_h$ in Alg. 1. Then, since under $B_{1,h} \cap B_{2,h} \cap E_{h-1}$ we have that $\beta_h \leq 7/(11c_T)$, $\eta_h \leq 3/(11c_T) \leq 3/22$, then, by applying Proposition 3 to $\nu_h$ w.r.t. $\eta_h, \beta_h$, we have

$$\nu_h \leq \eta_h + (1 + \eta_h)\beta_h \leq \left(\frac{3}{11} + \left(1 + \frac{3}{22}\right)\frac{7}{11}\right)\frac{1}{c_T} < \frac{1}{c_T}.$$

Then $\frac{1}{T} \leq \frac{1 - 2\nu_h}{1 - \nu_h}$ and $\frac{1}{1 - \nu_h} \leq \min(T, 2)$, so by applying Thm. 3, we have

$$\frac{1}{T}\widehat{\ell}(x, \lambda_h) \leq \widehat{\ell}_{J_h, \bar{A}_h}(x, \lambda_h) \leq \min(T, 2)\widehat{\ell}(x, \lambda_h).$$

**Analysis of $E_{h-1} \cap F_h$.** First note that under $E_{h-1}$ the following holds $\widehat{\ell}_{J_{h-1}, \bar{A}_{h-1}}(x, \lambda_{h-1}) \leq \min(T, 2)\widehat{\ell}(x, \lambda_{h-1})$, so, by applying Lemma 3 to $\widehat{\ell}_{J_{h-1}, \bar{A}_{h-1}}(x, \lambda_h)$, we have

$$d_h = \frac{n}{R_h}\sum_{u \in U_h} \widehat{\ell}_{J_{h-1}, \bar{A}_{h-1}}(x_u, \lambda_h) \leq \frac{\lambda_{h-1}n}{\lambda_h R_h}\sum_{u \in U_h}\widehat{\ell}_{J_{h-1}, \bar{A}_{h-1}}(x_u, \lambda_{h-1}) \leq \frac{2\lambda_{h-1}n}{\lambda_h R_h}\sum_{u \in U_h}\widehat{\ell}(x_u, \lambda_{h-1}).$$

Moreover under $F_h$, we have $\frac{n}{R_h}\sum_{u \in U_h}\widehat{\ell}(x_u, \lambda_{h-1}) \leq \max(5, \frac{6}{5}d_{\mathrm{eff}}(\lambda_{h-1}))$, so, under $E_{h-1} \cap F_h$, we have

$$d_h \leq 2q\max(5, (6/5)d_{\mathrm{eff}}(\lambda_{h-1})) \leq \max(10q, 3qd_{\mathrm{eff}}(\lambda_h)).$$

This implies that

$$|J_h| = M_h = q_2 d_h \leq q_2 \max(10q, 3qd_{\mathrm{eff}}(\lambda_h))$$

**Unrolling the recurrence relation.** The two results above imply $E_h \supseteq B_{1,h} \cap B_{2,h} \cap E_{h-1} \cap F_h$. Now we unroll the recurrence relation, obtaining

$$E_h \supseteq E_0 \cap (\cap_{j=1}^h F_j) \cap (\cap_{j=1}^h B_{1,j}) \cap (\cap_{j=1}^h B_{2,j}),$$

so by taking their intersections, we have

$$\cap_{h=0}^H E_h \supseteq E_0 \cap (\cap_{j=1}^H F_j) \cap (\cap_{j=1}^H B_{1,j}) \cap (\cap_{j=1}^H B_{2,j}). \tag{22}$$

**Bounding $B_{1,h}, B_{2,h}, F_h$ in high probability** Let $h \in [H]$. The probability of the event $B_{1,h}$ can be written as $\mathbb{P}(B_{1,h}) = \int \mathbb{P}(B_{1,h}|U_h, P_h) d\mathbb{P}(U_h, P_h)$. Now note that $\mathbb{P}(B_{1,h}|U_h, P_h)$ is controlled by Lemma 4, that proves that for any $U_h, P_h$, the probability of $\mathbb{P}(B_{1,h}|U_h, P_h)$ is at least $1 - \tau$. Then

$$\mathbb{P}(B_{1,h}) = \int \mathbb{P}(B_{1,h}|U_h, P_h) d\mathbb{P}(U_h, P_h) \geq \inf_{U_h} \mathbb{P}(B_{1,h}|U_h, P_h) \geq 1 - \tau.$$

To see that $\mathbb{P}(B_{1,h}|U_h, P_h)$ is controlled by Lemma 4, note that, since $|U_h|$ is exactly $R_h$, by definition of $\bar{A}_h$ and $A_h$

$$\bar{A}_h = \frac{|J_h|}{n} A_h = |U_h| \, \text{diag}(p_{j_1}, \dots, p_{j_{|J_h|}}),$$

that is exactly the condition on the weights required by Lemma 4 which controls exactly Equation (21). Finally $B_{2,h}, F_h$ are directly controlled respectively by Lemmas 5 and 6 and so hold with probability at least $1 - \tau$ each. Finally note that $E_0$ holds with probability 1. So by taking the intersection bound according to Equation (22), we have that $\cap_{h=0}^{H} E_h$ holds at least with probability $1 - 3H\tau$. $\qquad\square$

## A.6 Proof for Algorithm 2

**Lemma 7.** *Let $\lambda > 0$, $n \in \mathbb{N}$, $\delta \in (0, 1]$. Let $(x_i)_{i=1}^{n} \subseteq X$. Let $b \in (0, 1]$ and $p_1, \dots, p_n \in (0, b]$. Let $u_1, \dots u_n$ sampled independently and uniformly on $[0, 1]$. Let $v_j$ be independent $Bernoulli(p_j/b)$ random variables, with $j \in [n]$. Denote by $z_j$ the random variable $z_j = 1_{u_j \leq b} v_j$. Finally, let the random set $J$ containing $j$ iff $z_j = 1$. Let $A = \frac{n}{|J|}(p_{j_1}, \dots, p_{j_{|J|}})$, where $j_1, \dots, j_{|J|}$ are the sorting of $J$. Then the following holds with probability at least $1 - \delta$*

$$G_\lambda(\widehat{C}, \widehat{C}_{J,A}) \leq \frac{2s\eta}{3n} + \sqrt{\frac{2s\eta}{n}}, \quad with \quad s = \sup_{i \in [n]} \frac{1}{p_i} \|(\widehat{C} + \lambda I)^{-1/2} K_{x_i}\|_{\mathcal{H}}^2,$$

*with $s = \log \frac{4n}{\delta}$.*

*Proof.* Let $\zeta_i$ be defined as

$$\zeta_i = \frac{z_i}{p_i} \frac{1}{n} (\widehat{C} + \lambda I)^{-1/2} (K_{x_i} \otimes K_{x_i})(\widehat{C} + \lambda I)^{-1/2},$$

for $i \in [n]$, where $z_i$ are the Bernoulli random variables computed by Algorithm 2. First note that

$$(\widehat{C} + \lambda I)^{-1/2} \widehat{C}_{J,A} (\widehat{C} + \lambda I)^{-1/2} = \frac{1}{|J|} \sum_{j \in J} \frac{|J|}{np_j} (\widehat{C} + \lambda I)^{-1/2} (K_{x_i} \otimes K_{x_i})(\widehat{C} + \lambda I)^{-1/2}$$

$$= \frac{1}{n} \sum_{j \in J} \frac{1}{p_j} (\widehat{C} + \lambda I)^{-1/2} (K_{x_i} \otimes K_{x_i})(\widehat{C} + \lambda I)^{-1/2}$$

$$= \frac{1}{n} \sum_{i=1}^{n} \frac{z_i}{p_j} (\widehat{C} + \lambda I)^{-1/2} (K_{x_i} \otimes K_{x_i})(\widehat{C} + \lambda I)^{-1/2}$$

$$= \sum_{i=1}^{n} \zeta_i.$$

In particular we study the expectation and the variance of $\zeta_i$ to bound $G_\lambda(\widehat{C}, \widehat{C}_{J,A})$. By noting that the expectation of $z_i$ is $\mathbb{E}[z_i] = \mathbb{E}[1_{u_i \geq b} v_i] = \mathbb{E}[1_{u_i \geq b}]\mathbb{E}[v_i] = b \times \frac{p_i}{b} = p_i$, for any $i \in [n]$, then

$$\mathbb{E} \sum_{i=1}^{n} \zeta_i = \sum_{i=1}^{n} \frac{\mathbb{E}[z_i]}{p_i} \frac{1}{n} (\widehat{C} + \lambda I)^{-1/2} (K_{x_i} \otimes K_{x_i})(\widehat{C} + \lambda I)^{-1/2}$$

$$= \sum_{i=1}^{n} \frac{1}{n} (\widehat{C} + \lambda I)^{-1/2} (K_{x_i} \otimes K_{x_i})(\widehat{C} + \lambda I)^{-1/2}$$

$$= (\widehat{C} + \lambda I)^{-1/2} \widehat{C} (\widehat{C} + \lambda I)^{-1/2} =: W,$$

Now we will bound almost everywhere $\|\zeta_i\|$ as

$$\|\zeta_i\| \leq \sup_{i \in [n]} \frac{z_i}{p_i} \frac{1}{n} \|(\widehat{C} + \lambda I)^{-1/2} K_{x_i}\|_{\mathcal{H}}^2 \leq \frac{1}{n} \sup_{i \in [n]} \frac{1}{p_i} \|(\widehat{C} + \lambda I)^{-1/2} K_{x_i}\|_{\mathcal{H}}^2.$$

We are ready to apply non-commutative Bernstein inequality (Prop. 5 is specific version for this setting), obtaining, with probability at least $1 - \delta$

$$G_\lambda(\widehat{C}, \widehat{C}_{J,A}) = \|\frac{1}{n} \sum_{i=1}^n (\zeta_i - \mathbb{E}[\zeta_i])\| \leq \frac{2s\eta}{3n} + \sqrt{\frac{2s\eta}{n}},$$

with $\eta = \log \frac{4\,\mathrm{Tr}(W)}{\|W\|\delta}$. Finally note that since $\mathrm{Tr}(W)/\|W\| \leq \mathrm{rank}(W) \leq n$, we have $\eta \leq \log \frac{4n}{\delta}$. □

**Lemma 8.** *Let $\lambda > 0$, $n \in \mathbb{N}$, $\delta \in (0,1]$. Let $(x_i)_{i=1}^n \subseteq X$. Let $b \in (0,1]$ and $p_1, \ldots, p_n \in (0,b]$. Let $u_1, \ldots u_n$ sampled independently and uniformly on $[0,1]$. Let $v_j$ be independent $Bernoulli(p_j/b)$ random variables, with $j \in [n]$. Denote by $z_j$ the random variable $z_j = 1_{u_j \leq b} v_j$. Finally, let the random set $J$ containing $j$ iff $z_j = 1$. Then the following holds with probability at least $1 - \delta$*

$$|J| \leq \sum_{i \in [n]} p_i + (1 + \sqrt{\sum_{i \in [n]} p_i}) \log \frac{3}{\delta}.$$

*Proof.* By definition of $J_h$, note that

$$|J| = \sum_{i \in [n]} z_i.$$

We are going to concentrate the sum of random variables via Bernstein. Any $z_i$ is bounded, by construction, by 1. Moreover

$$\mathbb{E}[z_i] = \mathbb{E}[1_{u_i \geq b} v_i] = \mathbb{E}[1_{u_i \geq b}]\mathbb{E}[v_i] = b \times \frac{p_i}{b} = p_i.$$

Analogously $\mathbb{E}[z_i^2] - \mathbb{E}[z_i]^2 = p_i - p_i^2 \leq p_i$. By applying Bernstein inequality, we have

$$|\sum_{i \in [n]} (z_i - p_i)| \leq \log \frac{2}{\delta} + \sqrt{\log \frac{2}{\delta} \sum_{i \in [n]} p_i},$$

with probability $1 - \delta$. Then with the same probability,

$$|J| \leq \sum_{i \in [n]} p_i + (1 + \sqrt{\sum_{i \in [n]} p_i}) \log \frac{3}{\delta}.$$

□

**Theorem 5.** *Let $n \in \mathbb{N}$, $(x_i)_{i=1}^n \subseteq X$. Let $\delta \in (0,1]$, $t,q > 1$, $\lambda > 0$ and $H, d_h, \lambda_h, J_h, A_h$ as in Alg. 2. Let $\nu_h = G_\lambda(\widehat{C}, \widehat{C}_{J_h, \bar{A}_h})$. When*

$$\lambda_0 = \frac{\kappa^2}{\min(t,1)}, \quad q_1 \geq 2Tq(1 + 2/t) \log \frac{4n}{\delta}$$

*then, the following holds with probability $1 - \delta$: for any $h \in \{0, \ldots, H\}$*

$$a) \quad \frac{1}{T}\widehat{\ell}(x, \lambda_h) \leq \widehat{\ell}_{J_h, \bar{A}_h}(x) \leq \min(T,2)\widehat{\ell}(x, \lambda_h), \quad \forall x \in X,$$

$$b) \quad |J_h| \leq 3q_1 \min(T,2)(5 \vee d_{eff}(\lambda_h)) \log \frac{6H}{\delta}, \quad\quad (23)$$

$$c) \quad \nu_h \leq \frac{1}{c_T}.$$

*where $T = 1 + t$ and $c_T = 2 + 1/(T-1)$.*

*Proof.* Let $H, c_T, q$ and $\lambda_h, J_h, A_h, (p_{h,i})_{i=1}^n$ for $h \in \{0, \ldots, H\}$ as defined in Alg. 2 and define $\tau = \delta/(2H)$. Now we are going to define some events and we prove a recurrence relation that they satisfy. Finally we unroll the recurrence relation and bound the resulting events in probability.

**Definitions of the events**   Now we are going to define some events that will be useful to prove the theorem. Denote with $E_h$ the event such that the conditions in Eq. (23)-(a) hold for $J_h, \bar{A}_h$. Denote with $Z_h$ the event such that

$$|J_h| \leq \sum_{i \in [n]} p_{h,i} + (1 + (\sum_{i \in [n]} p_{h,i})^{1/2}) \log \frac{3}{\tau}.$$

Denote with $V_h$ the event such that $\nu_h := G_{\lambda_h}(\widehat{C}_{U,I}, \widehat{C}_{J_h, A_h})$, satisfies

$$\nu_h \leq s_h \log \frac{8\kappa^2}{\lambda_h \tau} + \sqrt{2 s_h \log \frac{8\kappa^2}{\lambda_h \tau}}, \quad \text{with} \quad s_h = \sup_{i \in [n]} \frac{1}{n p_{h,i}} \|(\widehat{C} + \lambda_h I)^{-1/2} K_{x_i}\|_{\mathcal{H}}^2. \quad (24)$$

**Analysis of $s_h$.**   Note that, by definition of $p_{h,i}$, for Algorithm 2, and of $\widehat{\ell}$, we have so

$$s_h = \sup_{i \in [n]} \frac{1}{n p_{h,i}} \|(\widehat{C} + \lambda_h I)^{-1/2} K_{x_i}\|_{\mathcal{H}}^2 = \sup_{i \in [n]} \frac{\widehat{\ell}(x_i, \lambda_i)}{q_1 \widetilde{\ell}_{J_h, A_h}(x_i)} = \sup_{i \in [n]} \frac{\widehat{\ell}(x_i, \lambda_i)}{q_1 \widehat{\ell}_{J_h, \bar{A}_h}(x_i)}.$$

with $\bar{A}_h = \frac{n}{|J|} A_h$, where the last step is due to the equivalence between $\widetilde{\ell}$ and $\widehat{\ell}$ in Proposition 1.

Now we are ready to prove the recurrence relation, for $h \in \{1, \ldots H\}$,

$$E_h \supseteq V_h \cap Z_h \cap E_{h-1}.$$

**Analysis of $E_0$.**   Note that, since $\|\widehat{C}\| \leq \kappa^2$, then $\frac{1}{\kappa^2 + \lambda} I \preceq (\widehat{C} + \lambda I)^{-1} \preceq \frac{1}{\lambda}$, so for any $x \in X$ the following holds

$$\frac{K(x,x)}{(\kappa^2 + \lambda)n} \leq \widehat{\ell}(x, \lambda) \leq \frac{K(x,x)}{\lambda n}.$$

Since $\lambda_0 = \frac{\kappa^2}{\min(2,T)-1}$ and $\widehat{\ell}_{\emptyset,[]}(x, \lambda_0) = \frac{K(x,x)}{\lambda_0 n}$, we have

$$\frac{1}{T} \widehat{\ell}(x, \lambda_0) \leq \frac{1}{T} \frac{K(x,x)}{\lambda n} \leq \ell_{\emptyset,[]}(x, \lambda_0) = \frac{K(x,x)}{\lambda_0 n} = \frac{\min(2,T) K(x,x)}{(\kappa^2 + \lambda_0)n} \leq \min(2,T) \widehat{\ell}(x, \lambda_0).$$

Setting conventionally $d_0, \nu_0, \eta_0, \beta_0 = 0$ (they are not used by the algorithm or the proof), we have that $E_0$ holds everywhere and so, with probability 1.

**Analysis of $E_{h-1} \cap V_h$.**   Note that under $E_{h-1}$, we have $\widehat{\ell}_{J_{h-1}, \bar{A}_{h-1}}(x, \lambda_{h-1}) \geq \frac{1}{T} \widehat{\ell}(x, \lambda_{h-1})$, so

$$s_h = \sup_{i \in [n]} \frac{\widehat{\ell}(x_i, \lambda_h)}{q_1 \widehat{\ell}_{J_h, \bar{A}_h}(x_i, \lambda_{h-1})} \leq T \sup_{i \in [n]} \frac{\widehat{\ell}(x_i, \lambda_h)}{q_1 \widehat{\ell}(x_i, \lambda_{h-1})}$$

$$\leq \frac{T \lambda_{h-1}}{\lambda_h} \sup_{i \in [n]} \frac{\widehat{\ell}(x_i, \lambda_{h-1})}{q_1 \widehat{\ell}(x_i, \lambda_{h-1})} = \frac{T \lambda_h}{q_1 \lambda_{h-1}} = \frac{Tq}{q_1},$$

where we used the fact that $\widehat{\ell}(x_i, \lambda_h) \leq \frac{\lambda_{h-1}}{\lambda_h} \widehat{\ell}(x_i, \lambda_{h-1})$, via Lemma 3. In particular since we are in $V_h$, this means that, since $q_1 \geq 2Tq(1 + 2/t) \log \frac{4n}{\delta}$, we have

$$\nu_h \leq \frac{Tq}{q_1} \log \frac{8\kappa^2}{\lambda_h \tau} + \sqrt{2 \frac{Tq}{q_1} \log \frac{8\kappa^2}{\lambda_h \tau}} \leq (4 + 2t^{-1})^{-2} + \sqrt{2/(4 + 2t^{-1})^2} \quad (25)$$

$$\leq (1/8 + \sqrt{1/8})(2 + t^{-1})^{-1} \leq \frac{1}{2c_T}. \quad (26)$$

Then $\frac{1}{T} \leq \frac{1 - 2\nu_h}{1 - \nu_h}$ and $\frac{1}{1 - \nu_h} \leq \min(T, 2)$, so by applying Thm. 3, we have

$$\frac{1}{T} \widehat{\ell}(x, \lambda_h) \leq \widehat{\ell}_{J_h, \bar{A}_h}(x, \lambda_h) \leq \min(T, 2) \widehat{\ell}(x, \lambda_h).$$

**Analysis of $E_{h-1} \cap Z_h$.** First consider $\sum_{i \in [n]} p_{h,i}$. By the fact that $\widetilde{\ell}_{J_{h-1}, A_{h-1}} = \widehat{\ell}_{J_{h-1}, \bar{A}_{h-1}}$, by Proposition 1, we have

$$
\sum_{i \in [n]} p_{h,i} = q_1 \sum_{i \in [n]} \widetilde{\ell}_{J_{h-1}, A_{h-1}}(x_i, \lambda_h) = q_1 \sum_{i \in [n]} \widehat{\ell}_{J_{h-1}, \bar{A}_{h-1}}(x_i, \lambda_h)
$$

$$
\leq q_1 \frac{\lambda_{h-1}}{\lambda_h} \sum_{i \in [n]} \widehat{\ell}_{J_{h-1}, \bar{A}_{h-1}}(x_i, \lambda_{h-1}), \leq q_1 \min(T, 2) \frac{\lambda_{h-1}}{\lambda_h} \sum_{i \in [n]} \widehat{\ell}(x_i, \lambda_{h-1}),
$$

$$
\leq q_1 \min(T, 2) \frac{\lambda_{h-1}}{\lambda_h} \sum_{i \in [n]} \widehat{\ell}(x_i, \lambda_h) = q_1 \min(T, 2) d_{\mathrm{eff}}(\lambda_h),
$$

where we applied in order (1) Lemma 3, to bound $\widehat{\ell}_{J_{h-1}, \bar{A}_{h-1}}(x_i, \lambda_h)$ in terms of $\widehat{\ell}_{J_{h-1}, \bar{A}_{h-1}}(x_i, \lambda_{h-1})$, (2) the fact that we are in the event $E_{h-1}$ and so $\widehat{\ell}_{J_{h-1}, \bar{A}_{h-1}}(x_i, \lambda_{h-1}) \leq \min(T, 2) \widehat{\ell}(x_i, \lambda_{h-1})$, then (3) again Lemma 3 to bound $\widehat{\ell}(x_i, \lambda_{h-1})$ w.r.t. $\widehat{\ell}(x_i, \lambda_h)$, and (4) finally the definition of $d_{\mathrm{eff}}(\lambda_h)$.

Now if $d_{\mathrm{eff}}(\lambda_h) \leq 10$, we have that

$$
\sum_{i \in [n]} p_{h,i} + (1 + (\sum_{i \in [n]} p_{h,i})^{1/2}) \log \frac{3}{\tau} \leq 15 q_1 \min(T, 2) \log \frac{3}{\tau}.
$$

If $d_{\mathrm{eff}}(\lambda_h) > 10$, we have that

$$
\sum_{i \in [n]} p_{h,i} + (1 + (\sum_{i \in [n]} p_{h,i})^{1/2}) \log \frac{3}{\tau} \leq 3 d_{\mathrm{eff}}(\lambda_h) q_1 \min(T, 2) \log \frac{3}{\tau}.
$$

So under $E_{h-1} \cap Z_h$, we have that

$$
|J| \leq 3 q_1 \min(T, 2) \left(5 \vee d_{\mathrm{eff}}(\lambda_h)\right) \log \frac{3}{\tau}.
$$

**Unrolling the recurrence relation.** The two results above imply $E_h \supseteq V_h \cap Z_h \cap E_{h-1}$. Now we unroll the recurrence relation, obtaining

$$
E_h \supseteq E_0 \cap (\cap_{j=1}^h Z_j) \cap (\cap_{j=1}^h V_j),
$$

so by taking their intersections, we have

$$
\cap_{h=0}^H E_h \supseteq E_0 \cap (\cap_{j=1}^H Z_j) \cap (\cap_{j=1}^H V_j). \tag{27}
$$

**Bounding $V_h, Z_h$ in high probability** Let $h \in [H]$. Denote by $P_h = (p_{h,j})_{j \in [n]}$. The probability of the event $Z_h$ can be written as $\mathbb{P}(Z_h) = \int \mathbb{P}(Z_h | P_h) d\mathbb{P}(P_h)$. Now note that $\mathbb{P}(Z_h | P_h)$ is controlled by Lemma 8, that proves that the probability of $\mathbb{P}(Z_h | P_h)$ is at least $1 - \tau$. Then

$$
\mathbb{P}(Z_h) = \int \mathbb{P}(Z_h | P_h) d\mathbb{P}(P_h) \geq \inf_{P_h} \mathbb{P}(Z_h | P_h) \geq 1 - \tau.
$$

The probability event $V_h$ is lower bounded by $1 - \tau$, via the same reasoning, using Lemma 7. Finally note that $E_0$ holds with probability 1. So by taking the intersection bound according to Equation (27), we have that $\cap_{h=0}^H E_h$ holds at least with probability $1 - 3H\tau$. $\qquad\square$

## A.7 Proof of Theorem 1

*Proof.* The proof of this theorem splits in the proof for Algorithm 1 that corresponds to Theorem 4 and the proof for Algorithm 2, that corresponds to Theorem 5. In particular, the result abou leverage scores is expressed in terms of out-of-sample-leverage-scores $\widehat{\ell}_{J_h, A_h}$ (Definition 1). The desired result, about $\widetilde{\ell}_{J_h, A_h}$, is obtained via Proposition 1.

Note that the two theorems provides stronger guarantees than the ones required by this theorem. We will use only points (a) and (b) of their statements. Moreover they prove the result for the out-of-sample-leverage-scores (Definition 1) and here we specify the result only for $x = x_i$, with $i \in [n]$. $\qquad\square$

# B Theoretical Analysis for Falkon with BLESS

In this section the FALKON algorithm is recalled in detail. Then it is proved in Thm. 6 that the excess risk of FALKON-BLESS is bounded by the one of Nyström-KRR. In Thm. 7 the learning rates for Nyström-KRR with BLESS are provided. In Thm. 8 a more general version of Thm. 2 is provided, taking into account more refined regularity conditions on the learning problem. Finally the proof of Thm. 2 is derived as a corollary.

## B.1 Definition of the algorithm

**Definition 2** (Generalized Preconditioner). *Given $\lambda > 0$, $(\widetilde{x}_j)_{j=1}^M \subseteq X$, $M \in \mathbb{N}$ and $A \in \mathbb{R}^{M \times M}$ positive diagonal matrix, we say that $B$ is a* generalized preconditioner, *if*

$$B = \frac{1}{\sqrt{n}} A^{-1/2} Q T^{-1} R^{-1},$$

*where $Q \in \mathbb{R}^{M \times q}$ partial isometry with $Q^\top Q = I$ and $q \leq M$, where $T, R \in \mathbb{R}^{q \times q}$ are invertible triangular, and $Q, T, R$ satisfy*

$$A^{-1/2} K_{MM} A^{-1/2} = Q T^\top T Q^\top, \quad R = \frac{1}{M} T T^\top + \lambda I,$$

*with $K_{MM} \in \mathbb{R}^{M \times M}$ defined as $(K_{MM})_{ij} = K(\widetilde{x}_i, \widetilde{x}_j)$.*

**Example 1** (Examples of Preconditioners). *The following are some ways to compute preconditioners satisfying Def. 2*

1. *If $K_{MM}$ in the definition above is full rank, then we can choose*

$$Q = I, \quad T = chol(A^{-1/2} K_{MM} A^{-1/2}), \quad R = chol(\frac{1}{M} T T^\top + \lambda I),$$

   *where chol is the Cholesky decomposition.*

2. *If $K_{MM}$ is rank deficient, let $q = rank(K_{MM})$, then*

$$(Q, Z) = qr(A^{-1/2} K_{MM} A^{-1/2}), \quad T = chol(Q^\top A^{-1/2} K_{MM} A^{-1/2} Q), \quad R = chol(\frac{1}{M} T T^\top + \lambda I),$$

   *where qr is the QR rank-revealing decomposition.*

3. *If instead of qr we want to use the eigendecomposition, then let $(\lambda_j, u_j)_{j=1}^M$ be the eigenvalue decomposition of $A^{1/2} K_{MM} A^{1/2}$ with $\lambda_1 \geq \cdots \geq \lambda_M \geq 0$ and let $q = rank(K_{MM})$. Then*

$$Q = (u_1, \ldots, u_q), \; T = diag(\sqrt{\lambda_1}, \ldots, \sqrt{\lambda_q}), \; R = diag\left(\sqrt{\frac{\lambda_1}{M} + \lambda}, \ldots, \sqrt{\frac{\lambda_q}{M} + \lambda}\right).$$

**Definition 3** (Generalized Falkon Algorithm). *Let $\lambda > 0$ and $t, n, M \in \mathbb{N}$. Let $(x_i, y_i)_{i=1}^n \subseteq X \times Y$ be the dataset. Given $J \subseteq [n]$ let $\widetilde{X}_J = \cup_{j \in J} x_j$ be the selected Nyström centers and denote by $\{\widetilde{x}_1, \ldots, \widetilde{x}_{|J|}\}$ the points in $\widetilde{X}_J$. Let $A \in \mathbb{R}^{|J| \times |J|}$ be a positive diagonal matrix of weights and $K$ the kernel function. Let $B, q$ be as in Def. 2 based on $\widetilde{X}_M$ and $A$. The* Generalized Falkon estimator *is defined as follows*

$$\widehat{f}_{\lambda,J,A,t} = \sum_{i=1}^{|J|} \alpha_i K(x, \widetilde{x}_i), \quad with \quad \alpha = B\beta_t,$$

*where $\beta_t \in \mathbb{R}^q$ denotes the vector resulting from $t$ iterations of the conjugate gradient algorithm applied to the following linear system*

$$W\beta = b, \quad W = B^\top (K_{nM}^\top K_{nM} + \lambda n K_{MM}) B, \quad b = B^\top K_{nM}^\top y,$$

*with $K_{nM} \in \mathbb{R}^{n \times M}$, $(K_{nM})_{ij} = K(x_i, \widetilde{x}_j)$, and $K_{MM} \in \mathbb{R}^{M \times M}$, $(K_{MM})_{ij} = K(\widetilde{x}_i, \widetilde{x}_j)$, and with $y = (y_1, \ldots, y_n) \in \mathbb{R}^n$.*

**Definition 4** (Standard Nyström Kernel Ridge Regression). *With the same notation as above, the standard Nyström Kernel Ridge Regression estimator is defined as*

$$\widetilde{f}_{\lambda,J} = \sum_{i=1}^{|J|} \alpha_i K(x, \widetilde{x}_i), \quad with \quad \alpha = (K_{nM}^\top K_{nM} + \lambda n K_{MM})^\dagger y.$$

## B.2 Main results

Here, Thm. 6 proves the excess risk of FALKON-BLESS is bounded by the one of Nyström-KRR. In Thm. 7 the learning rates for Nyström-KRR are provided. In Thm. 8 a more general version of Thm. 2 is provided, taking into account more refined regularity conditions on the learning problem. Finally the proof of Thm. 2 is derived as a corollary.

Let $Z_n = (x_i, y_i)_{i=1}^n$ be a dataset and $J \subseteq \{1, \ldots, n\}$ and $A \in \mathbb{R}^{|J| \times |J|}$ positive diagonal matrix. In the rest of this section we denote by $\widehat{f}_{\lambda,J,A,t}$ the Falkon estimator as in Def. 3 trained on $Z_n$ and based on the Nyström centers $\widetilde{X}_M = \cup_{j \in J} \{x_j\}$ and weights $A$ with regularization $\lambda$ and number of iterations $t$. Moreover we denote by $\widehat{f}_{\lambda,J}$ the standard Nyström estimator trained on $Z_n$ and based on the Nyström centers $\widetilde{X}_M$.

The following theorem is obtained by combining Lemma 2, 3 and Thm. 1 of [14], with our Prop. 2.

**Theorem 6.** *Let $\lambda > 0$, $n \geq 3$, $\delta \in (0, 1]$, $t_{\max} \in \mathbb{N}$. Let $Z_n = (x_i, y_i)_{i=1}^n$ be an i.i.d. dataset. Let $H$ and $(\lambda_h)_{h=0}^H$, $(M_h)_{h=0}^H$, $(J_h)_{h=0}^H$, $(A_h)_{h=0}^H$ be outputs of Alg. 1 runned with parameter $T = 2$.*

*The following holds with probability $1 - 2\delta$: for each $h \in \{0, \ldots, H\}$ such that $0 < \lambda_h \leq \|C\|$,*

$$\mathcal{R}(\widehat{f}_{\lambda_h,J_h,A_h,t}) \leq \mathcal{R}(\widetilde{f}_{\lambda_h,J_h}) + 4\widehat{v}\, e^{-t} \sqrt{1 + \frac{9\kappa^2}{\lambda_h n} \log \frac{nHt_{\max}}{\delta}}, \quad \forall t \in \{0, \ldots, t_{\max}\},$$

*with $\widehat{v} := \frac{1}{n} \sum_{i=1}^n y_i$.*

*Proof.* Let $\tau = \delta/(t_{\max} H)$ and let $h \in \{1, \ldots, H\}$. By Lemma 2 and Lemma 3 of [14], we have that, when $G_\lambda(\widehat{C}, \widetilde{C}_{J_h,A_h}) < 1$, with their $\widetilde{C}_{J_h,A_h} = \widehat{C}_{J_h,\bar{A}_h}$ and $\bar{A}_h$ defined as in theorem 4, then the condition number of $W_h$, that is the preconditioned matrix in Def. 3 with $\lambda = \lambda_h$, is controlled by

$$\mathrm{cond}(W_h) \leq \frac{1 + G_{\lambda_h}(\widetilde{C}_{J_h,A_h}, \widehat{C})}{1 - G_{\lambda_h}(\widetilde{C}_{J_h,A_h}, \widehat{C})}.$$

Now, by Prop. 2, we have

$$G_{\lambda_h}(\widetilde{C}_{J_h,A_h}, \widehat{C}) \leq \frac{G_{\lambda_h}(\widehat{C}, \widetilde{C}_{J_h,A_h})}{1 - G_{\lambda_h}(\widehat{C}, \widetilde{C}_{J_h,A_h})}.$$

So, combining the two results above, we have that when $G_{\lambda_h}(\widehat{C}, \widetilde{C}_{J_h,A_h}) \leq 1/3$

$$\mathrm{cond}(W_h) \leq \frac{1}{1 - 2\, G_{\lambda_h}(\widehat{C}, \widetilde{C}_{J_h,A_h})} \leq 3.$$

Now denote by $E_{h,t}$ the event such that

$$\mathcal{R}(\widehat{f}_{\lambda_h,J_h,A_h,t}) \leq \mathcal{R}(\widetilde{f}_{\lambda_h,J_h}) + 4\widehat{v}^2\, e^{-t} \sqrt{1 + \frac{9\kappa^2}{\lambda_h n} \log \frac{n}{\tau}}.$$

Since $\mathrm{cond}(W_h) \leq 3$, we have that $\log \frac{\sqrt{\mathrm{cond}(W_h)}+1}{\sqrt{\mathrm{cond}(W_h)}+1} \geq 1$ and so can apply Theorem 1 of [14] with their parameter $\nu = 1$, obtaining that each $E_{h,t}$, with $t \in \{0, \ldots, t_{\max}\}$ hold with probability $1 - \tau$. So by taking the intersection bound, we know that $E_h := \cap_{t=0}^{t_{\max}} E_{h,t}$ holds with probability $1 - t_{\max}\tau$.

Finally denote by $F_H$ the event: $G_{\lambda_h}(\widehat{C}, \widetilde{C}_{J_h,A_h}) \leq 1/3$ for any $h \in \{0, \ldots, H\}$. Note that Theorem 4 states that, by running Alg. 1 with $T = 2$, the event $F_H$ holds with probability at least $1 - \delta$.

The desired result correspond to the event $\cap_{h=1}^H E_h \cap F_H$ which, by taking the intersection bound, holds with probability at least $1 - \delta - t_{\max} H \tau$. $\qquad\square$

## B.3 Result for Nyström-KRR and BLESS

We introduce here the ideal and empirical operators that we will use in the following to prove the main results of this work and then we prove learning rates for Nyström-KRR.

In the following denote with $C : \mathcal{H} \to \mathcal{H}$ the linear operator

$$C = \int K_x \otimes K_x d\rho_X(x),$$

and, given a set of input-output pairs $\{(x_i, y_i)\}_{i=1}^n$ with $(x_i, y_i) \in X \times Y$ independently sampled according to $\rho$ on $X \times Y$, we define the empirical counterparts of the operators just defined as $\hat{S} : \mathcal{H} \to \mathbb{R}^n$ s.t.

$$f \in \mathcal{H} \mapsto \frac{1}{\sqrt{n}} (\langle K_{x_i}, f \rangle_{\mathcal{H}})_{i=1}^n \in \mathbb{R}^n,$$

with adjoint $\hat{S}^* : \mathbb{R}^n \to \mathcal{H}$ s.t.

$$v = (v_i)_{i=1}^n \in \mathbb{R}^n \mapsto \frac{1}{\sqrt{n}} \sum_{i=1}^n v_i K_{x_i},$$

Now we introduce some assumption that will be satisfied by the conditions on Thm. 2.

**Assumption 1.** *There exists $B, \sigma > 0$ such that the following holds almost everywhere on $X$*

$$\mathbb{E}[|y - \mathbb{E}[y|x]|^p \mid x] \leq \frac{p!}{2} B^{p-2} \sigma^2.$$

**Assumption 2.** *There exists $r \in [1/2, 1]$ and $g \in \mathcal{H}$ such that*

$$f_{\mathcal{H}} = C^{r-1/2} g,$$

**Theorem 7** (Generalization properties of Nyström-RR using BLESS). *Let $\delta \in (0, 1]$ and $\lambda > 0, n \in \mathbb{N}$. Under Asm. 1, 2, let the Nyström estimator as in Definition 4 and assume that $(J_h)_{h=1}^H, (A_h)_{h=1}^H, (\lambda_h)_{h=1}^H$ is obtained via Alg. 1 or 2. When $\frac{9\kappa^2}{n} \log \frac{n}{\delta} \leq \lambda \leq \|C\|$, then the following holds with probability $1 - 4\delta$*

$$\mathcal{R}(\widetilde{f}_{\lambda_h, J_h}) \leq 8\|g\|_{\mathcal{H}} \left( \frac{B \log \frac{2}{\delta}}{n\sqrt{\lambda_h}} + \sqrt{\frac{\sigma^2 d_{eff}(\lambda_h) \log \frac{2}{\delta}}{n}} + \lambda_h^{1/2+v} \right).$$

*Proof.* The proof consists in following the decomposition in Thm. 1 of [15], valid under Asm. 2 and using our set $J_h$ to determin the Nyström centers. First note that under Assumption 2, there exists a function $f_{\mathcal{H}} \in \mathcal{H}$, such that $\mathcal{E}(f_{\mathcal{H}}) = \inf_{f \in \mathcal{H}} \mathcal{E}(f)$ (see [16] and also [17, 18]). According to Thm. 2 of [15], under Asm. 2, we have that

$$\mathcal{R}(\widetilde{f}_{\lambda_h, J_h})^{1/2} \leq q(\underbrace{\mathcal{S}(\lambda_h, n)}_{\text{Sample error}} + \underbrace{\mathcal{C}(M_h)^{1/2+v}}_{\text{Computational error}} + \underbrace{\lambda_h^{1/2+v}}_{\text{Approximation error}}),$$

where $\mathcal{S}(\lambda, n) = \|(C + \lambda I)^{-1/2}(\hat{S}_n^* \hat{y} - \hat{C}_n f_{\mathcal{H}})\|$ and $\mathcal{C}(M_h) = \|(I - P_{M_h})(C + \lambda I)^{1/2}\|^2$ with $P_{M_h} = \hat{C}_{J_h, I} \hat{C}_{J_h, I}^{\dagger}$. Moreover $q = \|g\|_{\mathcal{H}}(\beta^2 \vee (1 + \theta\beta))$, $\beta = \|(\hat{C}_n + \lambda I)^{-1/2}(C + \lambda I)^{1/2}\|$, $\theta = \|(\hat{C}_n + \lambda I)^{1/2}(C + \lambda I)^{-1/2}\|$.

The term $\mathcal{S}(\lambda_h, n)$ is controlled under Asm. 1 by Lemma 4 of the same paper, obtaining

$$\mathcal{S}(\lambda, n) \leq \frac{B \log \frac{2}{\delta}}{n\sqrt{\lambda_h}} + \sqrt{\frac{\sigma^2 d_{\text{eff}}(\lambda_h) \log \frac{2}{\delta}}{n}},$$

with probability at least $1 - \delta$. The term $\beta$ is controlled by Lemma 5 of the same paper,

$$\beta \leq 2,$$

with probability $1 - \delta$ under the condition on $\lambda$. Moreover

$$\theta^2 = \|(C + \lambda I)^{-1/2} \hat{C}(C + \lambda I)^{-1/2}\| \leq 1 + \|(C + \lambda I)^{-1/2}(\hat{C} - C)(C + \lambda I)^{-1/2}\|,$$

where the last term is bounded by $1/2$ with probability $1 - \delta$ under the same condition on $\lambda$, via Prop. 8 and the following Remark 1 of the same paper.

Now we study the term $\mathcal{C}(M_h)$ that is the one depending on the result of BLESS. First note that, since $\text{diag}(A_h) > 0$, then

$$P_{M_h} = \widehat{C}_{J_h,I}\widehat{C}^{\dagger}_{J_h,I} = \widehat{C}_{J_h,\bar{A}_h}\widehat{C}^{\dagger}_{J_h,\bar{A}_h}.$$

By applying Proposition 3 and Proposition 7 of the same paper, the following holds

$$\mathcal{C}(M_h) \leq \frac{\lambda_h}{1 - G_{\lambda_h}(\widehat{C}, \widehat{C}_{J_h,\bar{A}_h})}, \leq 2\lambda_h,$$

with probability at least $1 - \delta$, where we applied Thm. 4-(c) and Thm. 5-(c), which control exactly $G_{\lambda_h}(\widehat{C}, \widehat{C}_{J_h,\bar{A}_h})$ and prove it to be smaller than $1/2$ in high probability.

Finally by taking the intersection bound of the events above, we have

$$\mathcal{R}(\widetilde{f}_{\lambda_h,J_h})^{1/2} \leq 4\|g\|_{\mathcal{H}} \left( \frac{B \log \frac{2}{\delta}}{n\sqrt{\lambda_h}} + \sqrt{\frac{\sigma^2 d_{\text{eff}}(\lambda_h) \log \frac{2}{\delta}}{n}} + 2\lambda_h^{1/2+v} \right),$$

with probability $1 - 4\delta$. $\qquad \square$

**Theorem 8** (Generalization properties of learning with FALKON-BLESS). *Let $\delta \in (0,1]$ and $\lambda > 0, n \geq 3, t_{\max} \in \mathbb{N}$. Let $Z_n = (x_i, y_i)_{i=1}^n$ be an i.i.d. dataset. Let $H$ and $M_H, J_H, A_H$ be outputs of Alg. 1 runned with parameter $T = 2$. Let $y \in [-a/2, a/2]$ almost surely, with $a > 0$. Under 2, Let $\lambda > 0$, $n \geq 3$, $\delta \in (0,1]$, when $\frac{9\kappa^2}{n} \log \frac{n}{\delta} \leq \lambda \leq \|C\|$, then the following holds with probability $1 - 6\delta$*

$$\mathcal{R}(\widehat{f}_{\lambda,J_H,A_H,t}) \leq 4a\, e^{-t} + 32\|g\|_{\mathcal{H}}^2 \left( \frac{a^2 \log^2 \frac{2}{\delta}}{n^2\lambda} + \frac{a d_{\text{eff}}(\lambda) \log \frac{2}{\delta}}{n} + 2\lambda^{1+2r} \right), \quad \forall t \in \{0, \ldots, t_{\max}\},$$

*Proof.* The result is obtained by combining Thm. 6, with Thm. 7 and noting that when $y \in [-a/2, a/2]$ almost surely, then it satisfies Asm. 1 with $B, \sigma \leq a$. $\qquad \square$

### B.4   Proof of Thm. 2

*Proof.* The result is a corollary of Thm. 8, where we assumed only the existence of $f_{\mathcal{H}}$. This correspond to assume Asm. 2, with $r = 1/2$ and $g = f_{\mathcal{H}}$ (see [16]). $\qquad \square$

## C   More details about BLESS and BLESS-R

**BLESS (Alg. 1).** Here we describe our bottom-up algorithm in detail (see Algorithm 1). The central element is using a decreasing list of $\{\lambda_h\}_{h=1}^h$, from a given $\lambda_0 \gg \lambda$ up to $\lambda$. The idea is to iteratively construct a LSG set that approximates well the RLS for a given $\lambda_h$, based on the accurate RLS computed using a LSG set for $\lambda_{h-1}$. The crucial observation of the proposed algorithm is that when $\lambda_{h-1} \geq \lambda_h$ then

$$\forall i : \ell(i, \lambda_h) \leq \frac{\lambda_h}{\lambda_{h-1}} \ell(i, \lambda_{h-1}), \qquad d_{\text{eff}}(\lambda_h) \leq \frac{\lambda_h}{\lambda_{h-1}} d_{\text{eff}}(\lambda_{h-1}),$$

(see Lemma 3, for more details). By smoothly decreasing $\lambda_h$, the LSG at step $h$ will only be a $\lambda_h/\lambda_{h-1}$ factor worse than our previous estimate, which is automatically compensated by a $\lambda_h/\lambda_{h-1}$ increase in the size of the LSG. Therefore, to maintain an accuracy level for the leverage scores approximation as in Eq. (2) and small space complexity, it is sufficient to select a logaritmically spaced list of $\lambda$'s from $\lambda_0 = \kappa^2$ to $\lambda$ (see Thm. 1), in order to keep $\lambda_h/\lambda_{h-1}$ as a small constant. This implies an extra multiplicative computational cost for the whole algorithm of only $\log(\kappa^2/\lambda)$.

More in detail, we initialize the Algorithm setting $D_0 = (\emptyset, [])$ to the empty LSG. Afterwards, we begin our main loop where at every step we reduce $\lambda_h$ by a $q$ factor, and then use $D_{h-1}$ to construct a new LSG $D_h$. Note that at each iteration we construct a set $J_h$ larger than $J_{h-1}$, which requires

computing $\widetilde{\ell}_{D_{h-1}}(i, \lambda_h)$ for samples that are not in $J_{h-1}$, and therefore not computed at the previous step. Computing approximate leverage scores for the whole dataset would be highly inefficient, requiring $\mathcal{O}(nM_h^2)$ time which makes it unfeasible for large $n$. Instead, we show that to achieve the desired accuracy it is sufficient to restrict all our operations to a sufficiently large intermediate subset $U_h$ sampled uniformly from $[n]$. After computing $\widetilde{\ell}_{D_{h-1}}(i, \lambda_h)$ only for points in $U_h$, we select $M_h$ points with replacements according to their RLS to generate $J_h$. With a similar procedure we update the weights in $A_h$. We will see in Thm. 1, $|U_h| \propto 1/\lambda_h$ is sufficient to guarantee that this intermediate step produces a set satisfying Equation (2), and also takes care of increasing $|U_h|$ to increase accuracy as $\lambda_h$ decreases. Moreover the algorithm uses a $M_h \propto \sum_{u \in U_h} \widetilde{\ell}_{D_{h-1}}(i, \lambda_h)$ that we prove in Thm. 1, to be in the order of $d_{\text{eff}}(\lambda_h)$. In the end, we return either the final LSG $D_H$ to compute approximations of $\ell(i, \lambda)$, or any of the intermediate $D_h$ if we are interested in the RLSs along the regularization path $\{\lambda_h\}_{h=1}^H$.

**BLESS-R (Alg. 2)** The second algorithm we propose, is based on the same principles of Algorithm 1, while simplifying some steps of the procedure. In particular it removes the need to explicitly track the normalization constant $d_h$ and the intermediate uniform sampling set, by replacing it with *rejection* sampling. At each iteration $h \in [H]$, instead of drawing the set $U_h$ from a uniform distribution, and then sampling $J_h$, from $U_h$, Algorithm 2 performs a single round of rejection sampling for each column according to the following identity

$$\mathbb{P}(z_{h,i} = 1) = \mathbb{P}(z_{h,i} = 1 | u_{h,i} \leq \beta_h)\mathbb{P}(u_{h,i} \leq \beta_h) = \beta_h p_{h,i}/\beta_h = p_{h,i} \propto \widetilde{\ell}_{D_{h-1}}(x_i, \lambda_{h-1}),$$

where $z_{h,i}$ is the r.v. which is 1 if $i \in [n]$, while $u_{h,i}$ is the probability that the column $i$ passed the rejection sampling step, while $\beta_h$ a suitable treshold which mimik the effect of the set $U_h$.

**Space and time complexity.** Note that at each iteration constructing the generator $\widetilde{\ell}_{D_{h-1}}$, requires computing the inverse $(K_{J_h} + \lambda_h nI)^{-1}$, with $M_h^3$ time complexity, while each of the $R_h$ evaluations $\widetilde{\ell}_{D_{h-1}}(i, \lambda_h)$ takes only $M_h^2$ time. Summing over the $H$ iterations Alg. 1 runs in $\mathcal{O}(\sum_{h=1}^H M_h^3 + R_h M_h^2)$ time. Noting that $R_h \simeq 1/\lambda_h$, that $M_h \simeq d_h \leq 1/\lambda_h$, and that $\sum_h \lambda_h^{-1} = \sum_h q^{h-H}\lambda^{-1} = \frac{q-q^{-H}}{q-1}\lambda^{-1}$, the final cost is $\mathcal{O}\left(\lambda^{-1} \max_h M_h^2\right)$ time, and $\mathcal{O}\left(\max_h M_h^2\right)$ space. Similarly, Alg. 2 only evaluates $\widetilde{\ell}_{D_{h-1}}$ for the points that pass the rejection steps which w.h.p. happens only $\mathcal{O}(n\beta_h) = \mathcal{O}(1/\lambda)$ times, so we have the same time and space complexity of Alg. 1.