[Reviews · NeurIPS 2018]

Reviewer 1



After Author Response: Thanks to the authors for their various clarifications. After further thought, however, I have updated my review to a 5. While this works seems to have very high potential, the current draft has several important shortcomings: (1) It is not yet polished enough for broad consumption. Significant revisions would be necessary to increase clarity. (2) The empirical section could be significantly improved; in particular, the experiments should be used to more directly validate the claims from the theorems, and to disentangle the issues related to faster optimization vs. improved final model --- this discussion should also be much more prominent. ===================================== Summary: This paper presents a novel algorithm for estimating leverage scores for kernel matrices. The running time of this algorithm improves upon state of the art. This algorithm is based on choosing samples of ever increasing size with which to approximate the leverage scores. These approximate leverage scores are then used to choose the Nystrom centers in the FALKON algorithm [1] (this method is named FALKON-LSG). The paper proves that given this choice of centers, FALKON-LSG achieves optimal learning rates (like FALKON does) but with improved runtime. It validates the quality of the estimated leverage scores, the runtime of their algorithm, and the accuracy of FALKON-LSG, empirically. The experiments show that FALKON-LSG converges faster than the original FALKON, though the two algorithms generally converge to very similar final test accuracies. I will now summarize the quality, clarity, originality, and significance of the work: 1) Quality: The general quality of the work seems high to me (though I have not checked the proofs). My primary concerns over the quality of the work are: -- There appear to be a fair number of careless mistakes throughout the paper, both in terms of the math and the text (these are listed at the end of the review). This makes me concerned that perhaps that proofs also have careless mistakes scattered throughout, which would suggest that the work isn't yet polished enough for publication. -- One of the primary theoretical claims of the paper is that this new algorithm for leverage score sampling allows for achieving the same generalization performance as FALKON and exact kernel ridge regression, but with improved running time. In particular, the claim appears to be that only O(d_eff) Nystrom centers need to be sampled (using approximate leverage scores) to achieve the same generalization performance as sqrt(n) Nystrom centers with the standard FALKON algorithm; given that d_eff is often significantly smaller than sqrt(n), this suggests that FALKON-LSG can use far fewer centers to achieve the same accuracy as FALKON. However, this claim is not validated empirically. In fact, the paper states that FALKON-LSG and standard FALKON converge to very similar test accuracy, using the same number of centers. This raises questions about when/why this algorithm would be of practical importance. It would have been informative to see a plot, with the x axis as the number of Nystrom centers, and the y axis as test performance (at convergence), for FALKON-LSG and FALKON. One thing the experiments *do show*, however, is that FALKON-LSG converges **faster** than FALKON; but this is a bit confusing, given that the rate of convergence was not something the theory addressed in any way. The paper does not dig into why this might be the case. It also shows FALKON-LSG is more robust to the choice of regularizer \lambda than FALKON, but this also isn't explained. 2) Clarity: If I were to ignore the list of small errors (in math and text) summarized below, I would say the paper is in general quite clear. But unfortunately, the presence of all these small issues made the paper somewhat laborious to get through. On the bright side, these issues could be fixed quite quickly. Some more minor comments: - Lines 137-138: I would have liked a more careful explanation of the time and space bounds. - In plots, legends and axis labels have tiny text. 3) Originality: The work is original, giving two new algorithms for leverage score sampling, and proving that these algorithms can achieve the same leverage score approximation error as existing algorithms, but can do so asymptotically faster. Furthermore, it presents a way of using these approximate leverage scores in the supervised learning setting (regression with least squares loss) in order to achieve optimal rates asymptotically faster than existing work. 4) Significance: From a theoretical perspective, the work is definitely significant. The paper presents the fastest known algorithm for achieving optimal learning rates for kernel ridge regression. However, it remains unclear what the practical significance of the work is, given that the paper was unable to show improved test performance over FALKON for a fixed number of Nystrom centers. [1] Alessandro Rudi, Luigi Carratino, Lorenzo Rosasco: FALKON: An Optimal Large Scale Kernel Method. NIPS 2017. ========================================================================== Mathematical issues found in paper: - Line 53: Isn't d_eff bounded above by n, not sqrt(n)? For example, when lambda is infinitesimal Tr(KK^(-1)) = n. - Line 61: The RLS of sample x_i seems to be an indication of how orthogonal phi(x_i) is, not x_i. I think this was meant as an informal statement, so not a huge deal. - Line 73: It says "corresponds exactly to \tilde{l}", and should say "corresponds exactly to l". - Line 77: |D| not defined. It seems |D| = |J|. - Algorithm 1: --- l(x_u, lambda) should be l(u,lambda) --- R_h as defined doesn't depend on h. I believe it should be R_h = q_1 min(\kappa^2/ \lambda_h, n). - Lines 89-90: The summary of lemma 3 is incorrect. Should be \lambda_h \leq \lambda_{h-1} \Rightarrow l(i, \lambda_h) \leq (\lambda_{h-1}/\lambda_h) l(i,\lambda_{h-1}). - Lines 101-102: Should be O(n M_{h-1}^2). I would then specify that this is made more efficient by only computing the approximate leverage scores for R_h points, which takes O(R_h M_{h-1}^2) time. - Algorithm 2: --- Failure probability \delta is taken as input but never used. --- Line 3: \lambda_h / q. What is q? Only q_1 and q_2 are defined as inputs. --- Line 4: q_1 q / (\lambda_h n). Once again, what is q? --- Line 6: uniform in what range? I assume [0,1] --- Line 8: Shouldn't it be min(\beta_h, ...), not min(1, ...)? This would ensure that p_{h,i}/\beta_h in [0,1]. - Lines 113-118. h-1 and h are confused in many places. computing l_{D_{h-1}} requires inverting (K_{J_{h-1}} + \lambda_{h-1}n I), which costs M_{h-1}^3 time, and each of the R_h evaluations takes M_{h-1}^2 time. So Algorithm 1 runs in time O( \sum_{h=1}^H M_{h-1}^3 + R_h M_{h-1}^2) time. - Lines 113-118: I found the derivation of the time and space bounds confusing. Shouldn't time bound be something like O(\sum_h \lambda_h^{-3}) (and then this should be simplified to be in terms of \lambda). - As written, algorithm 1 returns (M_h, J_h, A_h), and algorithm 2 returns (J_h, A_h) for h \in {1,...,H}. This inconsistency is weird. Also, it seems like the output of these algorithms should simply be (J_H,A_H), the LSG from the final iteration. This seems to be the way algorithm 3 uses the LSG algorithm in line 1. Another inconsistency is that Algorithm 2 takes as input a failure probability delta (which is never used), and Algorithm 1 doesn't. - Algorithm 3: --- Line 1: "(J,A) = LSG-Alg(X,\lambda,K) with approximation factor T = 2 and step q = 2". As mentioned above, (J,A) as output of the algorithm currently doesn't type check. Also, algorithms 1 and 2 require as input {q_1, q_2, \lambda_0}, not {T, q}. - line 172 and Eq (2): Should 'H' be used instead of 'h'? Issues in text: NIT - In many places, "leverage scores sampling" and "leverage scores generators" is written. It seems grammatically it should be "leverage score sampling" and "leverage score generators". - Two broken references on page 1 (lines 17 and 31). - Line 42: establish --> establishes - Line 44: "Our second, contribution..." --- comma after "second" is unnecessary. - Algorithm 2 title: Regection --> Rejection. - Line 112: mimik --> mimic. - Algorithm 3: broken equation reference "Eq. ??" - line 122: create --> creates. - line 160: "consists in a efficient" ---> "consists of an efficient". - line 190: "standard basic" --> "standard". - line 208: Broken theorem reference "Thm. ??" - line 209: "In Thm" -- Thm number not specified. - line 256: "Nystromcentres" --> "Nystrom centers". - Figures: --- Unit of time not specified in Figure 1. --- Figure 1 is actually a table (should be Table 2) --- The RLS accuracy box plot should be a separate figure with its own caption probably. --- Instead of calling the proposed algorithms "BLESS" and "BLESS-R" as these methods are called throughout the text, it says "Opt LS" and "REJLS". This is confusing. Also, RRLS is called Musco in two of the plots, which is also confusing.

Reviewer 2



After rebuttal: The authors addressed my minor concerns, and my score (accept) remains unchanged. The authors provide two successive overapproximation algorithms that efficiently sample from the rows of a kernel matrix according to an approximation of their leverage scores. They then provide a preconditioned ridge regression algorithm that uses these samples to efficiently approximately solve kernel ridge regression problems. The algorithms are refinements of previous algorithms published in AISTATS and NIPS, with superior dependence on the parameters of the problem. In particular, all three algorithms have run time that depend only on the effective dimensionality of the dataset, not the number of points in the training dataset. These are the first such algorithms, so the paper represents a significant contribution towards our understanding of the time complexity of solving kernel problems. The approach itself is also novel as far as I am aware: it is essentially a homotopy approach that samples a number of points that depends on the condition number of the dataset and the current regularization parameter, samples from these points using the current overestimate of the leverage scores, then uses the resulting points to estimate less overestimated leverage scores, and continues this process until the overestimation factor is 1. Strengths: - novel approach to estimating ridge leverage scores that may lead to more interesting work in the future - provide algorithms whose runtimes are independent of the size of the dataset - strong experimental evidence supports the claim that these algorithms are the current best available for these problems, measured both with respect to time and accuracy Weaknesses: - The paper is very dense, and hard to follow/digest. Rather than present to empirically and theoretically equivalent algorithms for leverage score sampling, I would prefer the authors present the rejection sampling algorithm which is simpler, and dedicate more time to explaining why it works intuitively - The paper needs thorough proofreading for readability, grammar, typos, inconsistent and wrong capitalization, bad and unexplained notation, and missing references to equations, theorems, and citations. As examples, I point to the two almost consecutive typos on line 112; the incorrect definition of the indicator variable on line 111, the unused input q_2 to Algorithm 2; the incorrect dependence on lambda instead of lambda_h in line 3 of Algorithm 1; the incorrect inequality on line 90; the dependence on lambda instead of lambda_h on line 108; the unexplained constant c2 on line 133; and the way that t is used in half of the paper to mean an accuracy parameter then it later becomes a number of iterations. - The experimental section uses the terms Opt LS and REJLS in the plots and tables, but refers to them as BLESS and BLESS-R in the text. Similarly it uses RRLS and "musco" interchangeably without specifying these are the same. The plots cannot be read when printed in black and white: please add markers and change the line styles, and use larger labels on the axes. Figures 4 and 5 would be more informative if they plotted the AUC vs time, as described in the text. - What is Nystrom-RR, referenced on line 189? What is the function g in Theorem 2?

Reviewer 3



Edit: My score remains the same after the rebuttal. They addressed my minor comments, but no additional experimental results are shown. Summary The paper proposed a novel sampling method to compute the approximated leverage score quickly and then use the sampled row and column based on the leverage score to compute the kernel ridge regression with computation complexity smaller than existing methods. Though existing approach to compute the approximated leverage score by first sample larger candidate set uniformly and then resample smaller set based on the first candidate set, the proposed method adopt a bottom-up approach that first much small subset and then gradually increasing the set using the leverage score to the previous candidate with \lambda decreasing. After the index of samples used is determined by the algorithm, then they apply the modified FALKON [10] algorithm called FALKON-LSG that use the score calculated in the previous algorithm to reweight the sample. They theoretically and experimentally demonstrated that their method can calculate leverage score faster than existing methods with similar approximation accuracy and their FALKON-LSG converges faster than FALKON with uniform sampling. Qualitative Assessment Their approach to calculate the leverage score and sampled index incrementally and use the calculated score for the approximate kernel ridge regression seems novel and rational. Our main concern is about the experiment. They fix \lambda in the experiment that measure the computation time (Figure 2). It seems unfair because the paper say to vary \lambda according to the number of samples. Thus, it seems better to change \lambda as N increases in Figure 2, too. I think the difference in computation time decreases in this setting because \lambda should be decrease as the number of samples grows. Also, as for the experiment for AUC, (Figure 4, 5) it is better to compare with other method SQUEAK and RRLS along with Uniform because uniform does not show good performance in figure 1. Some links to reference or others in the paper is corrupt. For example, L17, 31, 208 and Algorithm 3. In Algorithm 2, Rejection becomes Regection. Overall, except for the above faults, I did not find major drawback to counteract the potential impact of the paper. Thus, I vote for acceptance.

Reviewer 4



This paper presents a method for approximating leverage scores for kernel regression. They propose an algorithm which takes a bottom-up approach and incrementally enlarging the size of the set of columns. Then they argue that combining the leverage score sampling with a preconditioned conjugate gradient method achieves optimal generalization error in time complexity O(n d_eff), where d_eff is the effective dimension of the kernel learning problem, and previously best known running time for this problem is O(n d^2_eff). The presented algorithm and theoretical results are interesting. However, there are major presentation issues and omissions, especially in the numerical comparisons. Overall, the paper is worthy of publication based on technical contributions. Please see below for detailed comments. 1. The numerical comparisons don't seem to present a fair evaluation of the computational gains. When compared to uniform sampling, the authors only present comparisons within the FALCON algorithm (Figures 4 and 5). Uniform sampling can work better in practice when combined with other methods (e.g. Nystrom, direct solvers, CG and variants ). 2. The authors claim that the time complexity O(lambda^-1 d_eff) is independent in n, and the experiments employ a fixed value of lambda while n grows (Figure 1). However, as noted in Theorem 2, lambda must be set O(d_eff / n ) to achieve optimal generalization error. In this case, the complexity will depend on n. 3. What is 'OPT LS' in Figure 1 ? This doesn't appear in the text. 4. line 42, it's not clear what "state of the art accuracy" means since there is no mention of time complexity. Do you mean better accuracy in fixed time complexity? 5. line 44, typo 'our second,' 6. line 71, The notation for the empty subset \tilde l with double subscript looks odd, is this a typo ? 7. typo in Algorithm 2 description, 'rejection'